# AutoMS: Automatic Model Selection for Novelty Detection with Error Rate Control

**Yifan Zhang**[1*]   **Haiyan Jiang**[2,3†]   **Haojie Ren**[4†]   **Changliang Zou**[1]   **Dejing Dou**[3]

[1] School of Statistics and Data Sciences, LPMC, KLMDASR and LEBPS, Nankai University, Tianjin, China
[2] Machine Learning Department, MBZUAI, Abu Dhabi, UAE
[3] Baidu Research, Baidu Inc., Beijing, China
[4] School of Mathematical Science, Shanghai Jiao Tong University, Shanghai, China
`yfzhang_stat@mail.nankai.edu.cn, haiyan.jiang@mbzuai.ac.ae`
`haojieren@sjtu.edu.cn, zoucl@nankai.edu.cn, doudejing@baidu.com`

## Abstract

Given an unsupervised novelty detection task on a new dataset, how can we automatically select a "best" detection model while simultaneously controlling the error rate of the best model? For novelty detection analysis, numerous detectors have been proposed to detect outliers on a new unseen dataset based on a score function trained on available clean data. However, due to the absence of labeled anomalous data for model evaluation and comparison, there is a lack of systematic approaches that are able to select the "best" model/detector (i.e., the algorithm as well as its hyperparameters) and achieve certain error rate control simultaneously. In this paper, we introduce a unified data-driven procedure to address this issue. The key idea is to maximize the number of detected outliers while controlling the false discovery rate (FDR) with the help of Jackknife prediction. We establish non-asymptotic bounds for the false discovery proportions and show that the proposed procedure yields valid FDR control under some mild conditions. Numerical experiments on both synthetic and real data validate the theoretical results and demonstrate the effectiveness of our proposed AutoMS method. The code is available at: `https://github.com/ZhangYifan1996/AutoMS`.

## 1   Introduction

With the development of big data, novelty detection has become critical for many machine learning applications, such as network intrusion detection [1], fraud detection in finance [2, 3], event detection in the earth sciences [4], and so on. For these applications, there are a very large number of "normal" instances (i.e., inlier) but a lack of "abnormal" data (i.e., novelty, or outlier), because collecting abnormal data is much more expensive or difficult than collecting normal data [5]. This leads us to a situation where the available historical dataset contains only normal observations, and the basic goal is to detect outliers on a new unseen test dataset. Novelty detection has achieved remarkable success in many application domains with numerous methods/algorithms being developed in machine learning literature [6–8], however, two problems need to be addressed in the current novelty detection models, (1) lack of error rate control in detection algorithms, and (2) ad-hoc model selection of algorithms/hyperparameters.

The most common way to solve the novelty detection problem is to use a detection algorithm to learn a plausible score function that assigns a score to each point in the test dataset. Generally, data with large scores are more likely to be detected as outliers. A decision threshold is given to determine

---

[*]Y. Zhang did this work during her internship at Baidu Research.
[†]Corresponding author.

whether a data point is an outlier or not. There is an obvious trade-off between precision and recall: a small threshold would detect more outliers but make more false discoveries as well. Therefore, a more viable way is to reformulate the novelty detection problem as a multiple hypothesis testing problem, and adopt some error rate control procedure as suitable criteria to choose the threshold [9].

Model selection (including algorithm selection or hyperparameter tuning) is a well-known feature of these novelty detection algorithms and arises naturally since none of these methods can universally outperform others on *all* tasks as noted in Wolpert and Macready [10], Zhao et al. [11]. Due to the special setting of novelty detection, model evaluation and model comparison are not feasible when only "clean" data are provided as training dataset. Those strategies are usually ad-hoc [11, 12], which may greatly hamper their applicability to novelty detection in general cases. Furthermore, these methods only target the recall through model selection without considering the false discoveries. Therefore, it is desirable to develop a unified framework to select the "best" detector with certain error rate control.

We see that designing a data-driven procedure to select the "best" model/detector while simultaneously achieving some error rate control remains an open challenge. In this paper, we propose a data-driven procedure, AutoMS, to implement automatic model selection for novelty detection with error rate control. We apply the Jackknife technique [13] to fully explore the clean data and avoid the randomness caused by data-splitting. With the help of Jackknife prediction, we select from a pool of diversified models/detectors the "best" detector that maximizes the number of detected outliers while controlling the false discovery rate. The theoretical guarantees for the FDR control of the selected "best" detector requires further investigation because the error distribution of the "best" detector is different from that of a given detector. We address this issue by carefully studying the bound of false discovery proportion (FDP) and FDR which is calculated by taking expectation of FDP.

To our best knowledge, this is the first work to systematically select the "best" model while achieving certain error rate control. The main contributions of this work are summarized as follows.
(1) The proposed AutoMS can select the best model and simultaneously control the error rate of the best model. Notably, AutoMS is a unified data-driven framework, and it does not rely on any labeled anomalous data for model selection.
(2) To our best knowledge, it is the first effort to select the "best" model/detector with theoretical guarantees in the view of FDR control. We establish non-asymptotic bounds for the FDP and show that the proposed AutoMS yields valid FDR control.
(3) The AutoMS can be easily coupled with commonly used novelty detection algorithms. Extensive numerical experiments indicate that AutoMS outperforms other methods significantly, with respect to both error rate control and detection power.

## 2   Related Work

**FDR Control.**   Reformulating the outlier detection problem as a multiple testing problem, a commonly-used error rate in this setting is the false discovery rate (FDR): the expected proportion of true inliers among the test points reported as outliers. The FDR control [14–16] has been fully studied in the statistical literature and is a particularly useful tool to maintain the ability to reliably detect true alternatives without excessive false discovery results when large-scale hypotheses are simultaneously tested.

Along this line, Bates et al. [17] pioneered a method to split the clean dataset into two independent parts, one for training the score function and one for validating the empirical cumulative distribution function of the score function which will be used for constructing p-values for the new observations. They showed that the FDR control can be achieved with theoretical finite sample guarantees. It involves, however, a one-time random split of the clean dataset, which leads to a "p-value lottery" problem. That is, those p-values based on sample-splitting are sensitive and difficult to be reproduced in practice. Moreover, the procedure in Bates et al. [17] can only control the FDR for a given novelty detection model, and no model efficiency is considered in their paper.

**Model Selection in Novelty Detection.**   As described in Sect. 1, various detection algorithms have been developed for different problems or different types of data, such as LOF [18], one-class SVM [19], iForest [20] and so on. More discussions can be found in Pimentel et al. [6], Aggarwal [21]. However, no universal learning model performs well on all problem instances Wolpert and Macready [10]. Therefore, it raises the problem of model selection, including algorithm selection

and hyperparameter tuning. For example, AutoOD [12] proposed an automated outlier detection framework focusing on automatic neural architecture search. Recent work METAOD [11] built a measure for task similarity and use meta-learning to select the most similar task compared to the historical ones. Most existing work is usually based on sophisticated black-box machine learning algorithms to learn the space of the normal data. While generally effective in practice, they rarely offer a clear guarantee about the output.

We note that some statistical criteria can be used for model selection in novelty detection; e.g. using mass-volume and excess-mass [22] or some clustering validation metrics [23] to measure the quality of a score function, and using IREOS [24] to assess the separability. However, Ma et al. [25] indicated that those criteria are not useful in practice and cannot lead to a universal detector for general cases. Our proposed AutoMS is not limited to one specific algorithm or model but can select the "best" one among any set of models/detectors.

## 3 Methodology: Automatic Model Selection

In novelty detection problems, we are given a *clean* training dataset of inliers $\mathcal{D} = \{\tilde{Z}_j\}_{j=1}^{m}$ containing $m$ independently and identically distributed (i.i.d.) observations $\tilde{Z}_j$ which is drawn $\mathbb{P}_0$. The goal of the novelty detection problem is to identify the outlier set $\mathcal{O} = \{Z_i \in \mathcal{U} : Z_i \nsim \mathbb{P}_0\}$ among a new unseen test dataset $\mathcal{U} = \{Z_i\}_{i=1}^{n}$.

One of the most common approaches in addressing novelty detection is to employ a detection algorithm $\mathcal{M}$ to learn a plausible score function $S_{\mathcal{M}}(\cdot)$ with the clean dataset $\mathcal{D}$ and use the score $S_{\mathcal{M}}(Z_i)$ to assess whether $Z_i \in \mathcal{U}$ is an outlier. Generally, it is assumed that the larger the score $S_{\mathcal{M}}(Z_i)$, the more likely $Z_i$ is an outlier. Hence, the decision set (detected outlier set) is selected as: $\widehat{\mathcal{O}}_{\mathcal{M}} = \{Z_i \in \mathcal{U} : S_{\mathcal{M}}(Z_i) \geq L_{\mathcal{M}}\}$, where $L_{\mathcal{M}}$ is a threshold associated with detector $\mathcal{M}$. Denote the subset of $\mathcal{U}$ containing all inliers as $\mathcal{O}^c = \{Z_i \in \mathcal{U} : Z_i \sim \mathbb{P}_0\}$ with size $n_0$, and $\mathcal{O} \bigcup \mathcal{O}^c = \mathcal{U}$. Let $|\mathcal{A}|$ denote the number of the elements in set $\mathcal{A}$. Denote $a \vee b = \max\{a, b\}$.

Taking a multiple testing perspective, the null hypothesis is that a new observation is an inlier, and the alternative asserts it is an outlier. That is, the novelty detection problem is translated into a multiple testing problem as follows,

$$\mathbb{H}_{0i} : Z_i \sim \mathbb{P}_0 \text{ v.s. } \mathbb{H}_{1i} : Z_i \nsim \mathbb{P}_0 \quad i = 1, \cdots, n. \tag{1}$$

### 3.1 Problem Definition

**Problem (Model Selection of Unsupervised Novelty Detection)** The model selection problem for unsupervised novelty detection problem can be stated as follows.

*Given* a training dataset of inliers $\mathcal{D} = \{\tilde{Z}_j\}_{j=1}^{m}$, a new unlabeled dataset $\mathcal{U} = \{Z_i\}_{i=1}^{n}$ which may contain outliers, and all models $\mathcal{G} = \bigcup \mathcal{M}$;

*Select* the best model $\mathcal{M}^* \in \mathcal{G}$, *such that* the FDR of $\mathcal{M}^*$ is controlled under the target level $\alpha$.

The false discovery proportion (FDP) and true discovery proportion (TDP) are defined as follows

$$\text{FDP} = \frac{|\widehat{\mathcal{O}}_{\mathcal{M}} \bigcap \mathcal{O}^c|}{1 \vee |\widehat{\mathcal{O}}_{\mathcal{M}}|}, \quad \text{TDP} = \frac{|\widehat{\mathcal{O}}_{\mathcal{M}} \bigcap \mathcal{O}|}{|\mathcal{O}|}. \tag{2}$$

The false discovery rate (FDR) is the expectation of FDP, i.e., $\text{FDR} = \mathbb{E}(\text{FDP})$. The true detection rate (TDR) is defined as the expectation of TDP, i.e., $\text{TDR} = \mathbb{E}(\text{TDP})$. A detector is superior to another if the former has a larger TDR for the same FDR level.

### 3.2 Construction of AutoMS

We begin with the FDR control for a given detector $\mathcal{M}$ and then give the AutoMS of how to select the "best" detector from $\mathcal{G}$. For a detector $\mathcal{M}$, we learn a score function $S_{\mathcal{M}}(\cdot)$ based on the historical clean dataset $\mathcal{D}$ and evaluate the score $S_{\mathcal{M}}(Z_i)$ at the new data point $Z_i \in \mathcal{U}$. The canonical approach to FDR control in multiple testing problem is to apply the Benjamini-Hochberg (BH) procedure [14–16] to the p-values of scores $S_{\mathcal{M}}(Z_i)$ for any $Z_i \in \mathcal{U}$. That is, the data-dependent threshold $L$ is

selected as follows,

$$L = \inf \left\{ t > 0 : \frac{nG_{\mathcal{M}}(t)}{|\{Z_i \in \mathcal{U} : S_{\mathcal{M}}(Z_i) \geq t\}| \vee 1} \leq \alpha \right\},$$

where $G_{\mathcal{M}}(t) = \mathbb{P}_{\mathbb{H}_0}(S_{\mathcal{M}}(Z_i) \geq t | \mathcal{D})$ is the p-value of $S_{\mathcal{M}}(Z_i)$ for $Z_i \in \mathcal{U}$ when $\mathbb{H}_0$ holds.

However, it is quite a challenge to estimate reliable p-values $G_{\mathcal{M}}(t)$ since $\mathbb{P}_0$ is unknown and the score function $S_{\mathcal{M}}(\cdot)$ may be derived from a sophisticated black-box machine learning algorithm. Bates et al. [17] suggested randomly splitting $\mathcal{D}$ into two disjoint parts with equal sample size, one for training the score function $S_{\mathcal{M}}(\cdot)$ and the other for estimating $G_{\mathcal{M}}(t)$ with the help of conformal inference. This single-random-splitting based method suffers from, (1) the estimation of the p-values depends on the random split; (2) different split ratios also lead to different results.

To yield more stable estimates of p-values in practice, we propose to learn the score function $S_{\mathcal{M}}(\cdot)$ on the whole clean dataset $\mathcal{D}$, where our key idea stems from the **Jackknife estimate**. Denote $\mathcal{D}^{[-j]} = \mathcal{D} \backslash \{\tilde{Z}_j\}$ as the subset of $\mathcal{D}$ with the $j$-th observation removed. Define $S_{\mathcal{M}}^{[-j]}(\cdot)$ as the score function trained on $\mathcal{D}^{[-j]}$. Then we can evaluate $m$ scores $\{S_{\mathcal{M}}^{[-j]}(\tilde{Z}_j)\}_{j=1}^m$ on the whole clean dataset $\mathcal{D}$. The p-value function can be estimated by

$$G_{m,\mathcal{M}}(t) = \frac{|\{\tilde{Z}_j \in \mathcal{D} : S_{\mathcal{M}}^{[-j]}(\tilde{Z}_j) \geq t\}|}{m}. \tag{3}$$

The Jackknife technique provides an approximation of the score, $S_{\mathcal{M}}^{[-j]}(\tilde{Z}_j)$, for any $\tilde{Z}_j \in \mathcal{D}$. Under the null $\mathbb{H}_0$ that $Z_i \in \mathcal{U}$ is from $\mathbb{P}_0$, $S_{\mathcal{M}}^{[-j]}(\tilde{Z}_j)$ can be used for estimating the empirical distribution of $S_{\mathcal{M}}(Z_i)$. And the corresponding threshold $L_{\mathcal{M}}$ is determined via

$$L_{\mathcal{M}} = \inf \left\{ 0 < t < \bar{t}_{m,\mathcal{M}} : \frac{nG_{m,\mathcal{M}}(t)}{|\{Z_i \in \mathcal{U} : S_{\mathcal{M}}(Z_i) \geq t\}| \vee 1} \leq \alpha \right\}, \tag{4}$$

where $\bar{t}_{m,\mathcal{M}}$ is the upper bound for $t$ and its choice will be discussed in Sect. 4 in detail, and $S_{\mathcal{M}}(\cdot)$ is the same score function trained on $\mathcal{D}$. Then, the detected set is identified as

$$\widehat{\mathcal{O}}_{\mathcal{M}} = \{Z_i \in \mathcal{U} : S_{\mathcal{M}}(Z_i) \geq L_{\mathcal{M}}\}. \tag{5}$$

Further, we select the "best" detector $\mathcal{M}^*$ via

$$\mathcal{M}^* = \arg\max_{\mathcal{M} \in \mathcal{G}} |\widehat{\mathcal{O}}_{\mathcal{M}}|. \tag{6}$$

---

**Algorithm 1** AutoMS: Automatic Model Selection

1: **Input:** New dataset $\mathcal{U}$, clean dataset $\mathcal{D}$, target FDR level $\alpha$ and a pool of detectors $\mathcal{G}$.
2: **for** $\mathcal{M} \in \mathcal{G}$ **do**
3:     **for** $j = 1, \cdots, m$ **do**
4:         Train score function $S_{\mathcal{M}}^{[-j]}(\cdot)$ based on $\mathcal{D}^{[-j]}$ (i.e. $\mathcal{D}^{[-j]} = \mathcal{D} \backslash \{\tilde{Z}_j\}$ );
5:         Compute the score at $\tilde{Z}_j \in \mathcal{D}$, i.e., $S_{\mathcal{M}}^{[-j]}(\tilde{Z}_j)$;
6:     **end for**
7:     Learn score function $S_{\mathcal{M}}(\cdot)$ based on $\mathcal{D}$;
8:     Compute the scores $S_{\mathcal{M}}(Z_i)$ for any $Z_i \in \mathcal{U}$, and find a threshold $L_{\mathcal{M}}$ by Eq. (4);
9:     The detected set is $\widehat{\mathcal{O}}_{\mathcal{M}} = \{Z_i \in \mathcal{U} : S_{\mathcal{M}}(Z_i) \geq L_{\mathcal{M}}\}$.
10: **end for**
11: Select the "best" detector $\mathcal{M}^*$ from $\mathcal{G}$ via Eq. (6).
12: **Output:** $\widehat{\mathcal{O}}_{\mathcal{M}^*} = \{Z_i \in \mathcal{U} : S_{\mathcal{M}^*}(Z_i) \geq L_{\mathcal{M}^*}\}$.

---

We refer to our procedure as AutoMS short for *Automatic Model Selection* and it is summarized in Algorithm 1. In fact, our AutoMS consists of two phases: offline training in lines 2-7 and online prediction in lines 8-9 of pseudo-code in Algorithm 1. Note that the running time of offline training is not critical. Meanwhile, the additional running time is the detector selection in line 11, which should incur a small run-time overhead due to only maximizing a finite set.

Note that the pool of detectors $\mathcal{G} = \bigcup \mathcal{M}$ can be a set of different algorithms or a set of hyperparameters for one typical algorithm, or both, which fits the definition of model selection including

algorithm selection and hyperparameter tuning. Note that the size of $\mathcal{G}$, $|\mathcal{G}|$, can be infinite, and $|\mathcal{G}|$ is allowed to increase as the size of $\mathcal{D}$. More details will be discussed in Sect. 4.

**Rationale.** Here is the intuition why the selected detector obtained by AutoMS is the "best" one. The general idea is as follows. A good detector needs to achieve a large TDR while controlling the false discoveries under the target FDR level $\alpha$. **With FDR control**, the selected detector which has the largest number of discoveries is roughly the one with the largest TDR. Otherwise, simply selecting the largest number of discoveries **without FDR control**, is not even a correct criterion, because in this case, most detected discoveries may be inliers and true outliers may not be detected. For every detector $\mathcal{M} \in \mathcal{G}$, with FDR control, its FDP can be approximately controlled around the target FDR level $\alpha$, as the proposed AutoMS resets the corresponding novelty threshold for every $\mathcal{M}$. Therefore, there are roughly $|\widehat{\mathcal{O}}_{\mathcal{M}}|(1-\alpha)$ true outliers in the detected set $\widehat{\mathcal{O}}_{\mathcal{M}}$, with the TDR being approximated by $\frac{|\widehat{\mathcal{O}}_{\mathcal{M}}|(1-\alpha)}{|\mathcal{O}|}$. Since the number of true outliers $|\mathcal{O}|$ and the target FDR level $\alpha$ are fixed in a given novelty detection task, selecting the largest $\frac{|\widehat{\mathcal{O}}_{\mathcal{M}^*}|(1-\alpha)}{|\mathcal{O}|}$, namely the largest $|\widehat{\mathcal{O}}_{\mathcal{M}^*}|$, leads to the "best" detector $\mathcal{M}^*$.

**Why Jackknife not Jackknife+.** Barber et al. [26] introduced the Jackknife+ as a new method for constructing predictive confidence intervals. They provide non-asymptotic coverage guarantees under no assumptions beyond the training and test dataset being exchangeable and also study the theoretical guarantees for the original Jackknife with stability assumptions (see Section 5 of their paper). Jackknife+ needs to calculate $m$ leave-one-out predictions for one test point while the original Jackknife only calculates one predicted value [26]. The model evaluation cost (counts the number of times evaluating a fitted model on a new point) of Jackknife+ is $mn$ whereas the original Jackknife is of cost $n$, indicating that Jackknife+ has a much higher computation complexity compared to the original. Thus, since most novelty detection algorithms satisfy the stability assumption, we are more inclined to apply the original Jackknife technique rather than Jackknife+ to detectors that meet stability assumption, and we also give theoretical guarantees to achieve the non-asymptotic bounds of FDP and maintain FDR control.

## 4 Statistical Performance Guarantees

In this section, we will theoretically establish NON-asymptotic bounds for FDP of the selected detector $\mathcal{M}^*$ by AutoMS and further show that it can control the FDR under the following mild assumptions. Define the number of detectors $\varpi_m = |\mathcal{G}|$ and let $A_n = o(n)$ be a pre-specified sequence.

**Assumption 1** (Density). For any detector $\mathcal{M} \in \mathcal{G}$ and sufficiently large $m$, the conditional density of $S_{\mathcal{M}}(Z)$ given training dataset $\mathcal{D}$ is uniformly bounded above by a universal constant $c_f > 0$, where $Z \in \mathcal{U}$ is a new point.

**Assumption 2** (Learning stability). For any fixed $\mathcal{M}$, the score $S_{\mathcal{M}}$ satisfies: for large enough $m$,
$$\mathbb{P}(|S_{\mathcal{M}}(Z) - \tilde{\mu}_{\mathcal{M}}(Z)| \leq \kappa_m) \geq 1 - \zeta_m,$$
with some sequences satisfying $\kappa_m = o(1)$ and $\zeta_m = o(1)$ as $m \to \infty$, and some function $\tilde{\mu}_{\mathcal{M}}$.

**Assumption 3** (Rates). Denote $B_m = 2c_f\kappa_m + 3\zeta_m$. Suppose that $B_m = o(1)$, $n\varpi_m B_m = o(1)$ and $m\varpi_m/A_n^{2/3} = o(1)$.

Our main theoretical result on the validity of the AutoMS method for both FDP and FDR control is given by the next theorem.

**Theorem 1** (FDR control). *Suppose Assumptions 1–3 hold. Let $0 < \delta < 1$ and $0 < \alpha < 1$. There exist universal constants $C_1 > 0$ and $C_2 > 0$ so that the FDP of the proposed method satisfies*

$$\mathrm{FDP}(L_{\mathcal{M}^*}) \leq \alpha \left[ 1 + \frac{4nB_m}{A_n} + C_1\sqrt{\frac{\varpi_m}{\delta A_n^{2/3}}} + C_2\sqrt{\frac{\varpi_m W_{mn}}{\delta}} \right], \tag{7}$$

*with probability at least $1 - 2\delta$, and*

$$\limsup_{(m,n)\to\infty} \mathrm{FDR}(L_{\mathcal{M}^*}) \leq \alpha, \tag{8}$$

*where $W_{mn} = \left( \frac{n}{mA_n^{2/3}} + \frac{n}{A_n^{4/3}} \right)(1 + 15B_m + 2m^2B_m)$.*

It is worth mentioning that even if all detectors control the FDR, we cannot assert without theoretical proof that the optimal detector $\mathcal{M}^*$ can still guarantee the FDR control. And Bates et al. [17] only gives **the expectation version** that FDR can be controlled for a given detector. Briefly, we first prove that the FDP of each detector can be bounded *uniformly* through the uniform convergence of the Jackknife estimator. Hence, the selected "best" detector $\mathcal{M}^*$ with the largest number of discoveries will have the same non-asymptotic bounds for FDP. Then we give the asymptotic FDR control for the optimal detector. Detailed proofs for theoretical results are provided in **Appendix E**.

In Eq. (4), we specify a data-driven upper bound $\bar{t}_{m,\mathcal{M}}$ when determining $L_{\mathcal{M}}$. We define $\bar{t}_{m,\mathcal{M}} = G_{m,\mathcal{M}}^-(A_n/n)$ where $G_{m,\mathcal{M}}(t)$ is defined in Eq. (3) and $G_{m,\mathcal{M}}^-(y) = \inf\{t \geq 0 : G_{m,\mathcal{M}}(t) \leq y\}$ with $0 \leq y \leq 1$. The upper bound $\bar{t}_{m,\mathcal{M}}$ for threshold $L_{\mathcal{M}}$ can facilitate the derivation of non-asymptotic bounds for FDP but is actually not required in practice. In other words, this makes an implicit assumption that a non-trivial (not $\infty$) threshold can be attained within the interval $(0, \bar{t}_{m,\mathcal{M}}]$. By Theorem 1, we can see that a large $A_n$ would yield faster convergence of FDP. However, too large $A_n$ may lead to a trivial threshold. In fact, how large $A_n$ could be depends on the number of identifiable outliers in $\mathcal{U}$ [27], which is certainly unknown to us. Alternatively, by examining the requirements in Assumption 3, we can set $A_n \sim \min\{(m\varpi_m)^2, n^{1/2}\}$ for simplicity.

*Remark* 1 (Stability). Since we use the Jackknife to estimate FDP, certain stability conditions seem to be necessary. In Assumption 2, $\tilde{\mu}$ needs not to be close to an optimal population detector. We only need the $S_{\mathcal{M}}$ to concentrate around $\tilde{\mu}$. This is just a stability assumption rather than a consistency assumption. When the sampling stability fails to hold, the $S_{j,\mathcal{M}}^{[-j]}$ may have a substantially different distribution than $G_{n,\mathcal{M}}(t)$. In that case, we may resort to the sample-splitting procedure instead.

*Remark* 2 (Rates). In general, Assumption 3 is used to ensure the uniform consistency of the estimated p-values, which performs an important role in the convergence of the FDR control. Specifically, $A_n = o(n)$ is a pre-specified sequence which gives a non-trivial lower bound of the true p-value function $G_{\mathcal{M}}(t)$ with $G_{\mathcal{M}}(t) \geq \frac{A_n}{n} = o(1)$. The $B_m$ is a quantity used to measure the difference of two p-values estimated in different ways, as in Lemma C.3 and Lemma C.2. And $B_m = o(1)$ indicates the two p-values should be close enough which is a very mild condition, and it can be expected that the p-values will be close if the training sample sizes of inliers $m$ and test sample sizes of inliers $n_0$ are large enough in practice. In the proposed AutoMS, the number of detectors $\varpi_m = |\mathcal{G}|$ can go to infinity with some certain rate related to the training sample size $m$ and test sample size $n$. The assumptions $n\varpi_m B_m = o(1)$ and $m\varpi_m / A_n^{2/3} = o(1)$ indicate the required rate of $\varpi_m$ to ensure the uniform consistency among detectors.

*Remark* 3 (FDP bound). Although the FDP bound might be slightly larger than the target FDR level $\alpha$ in finite samples, the FDP bound converges asymptotically to the target FDR level $\alpha$. Actually, in novelty detection tasks, it is not necessary to obtain procedures of finite FDR control with additional assumptions, instead, a powerful and stable procedure which controls FDR asymptotically is good enough for most cases.

## 5 Experiments and Evaluation

### 5.1 Experiment Setting

In this section, we evaluate the performance of the proposed AutoMS in terms of FDR control and detection power (TDP) through extensive numerical studies on both synthetic and real data. We report the results based on 100 replication experiments throughout this paper. We construct a detector set $\mathcal{G}$ with 83 candidate models by combining 6 novelty detection algorithms and their corresponding hyperparameters (See Table S1 in **Appendix A** for the complete list). These models/detectors are available in the Python "**PyOD**" [28] package. Note that the size of detector set $\mathcal{G}$ can be infinite and our AutoMS can still be used.

**Performance Measures.** Since we focus on both error rate control and detection power, we use $\text{FDR} = \mathbb{E}(\text{FDP})$ and $\text{TDR} = \mathbb{E}(\text{TDP})$ as performance measures for each model in this paper. According to the definitions in Eq. (2) of Sect. 3, we have $\text{FDP} = 1 - \text{Precision}$, and $\text{TDP} = \text{Recall}$. Correspondingly, the empirical FDR and empirical TDR are calculated by averaging of FDPs and TDPs over 100 replications, respectively. We say a detection procedure is superior to its competitor if it has a larger TDR for the same target FDR level $\alpha$, that is, a superior detector has a larger empirical TDR whose empirical FDR is below or around the target FDR level $\alpha$. The preset target FDR level $\alpha$

should be an acceptable level, which means all detectors that control the FDR at this level $\alpha$ should be accepted. As long as the FDR of the selected detector is below a preset level $\alpha$, we have achieved our goal on FDR control and we should focus on improving TDR.

**Baseline methods.** To evaluate the performance of the proposed **AutoMS** method, we compare the proposed AutoMS with two different types of baselines: (i) model selection methods (METAOD [11]) and (ii) error-rate-controlled methods (SRS-based methods [17]). And we also consider two versions of the proposed AutoMS:

Table 1: Model-selection and error-rate-control methods.

| Method | Model Selection | Error Rate Control |
|---|---|---|
| SRS [17] | ✗ | ✓ |
| METAOD [11] | ✓ | ✗ |
| AutoMS-SRS | ✓ | ✓ |
| AutoMS-JK | ✓ | ✓ |

**AutoMS-JK** and **AutoMS-SRS**, where AutoMS-JK uses Jackknife technique and AutoMS-SRS replaces the Jackknife technique with single-random-splitting (SRS) technique for estimating score. We explore the performances of the four methods with respect to FDR and TDR. It is worth mentioning that **SRS** is guaranteed with FDR control for any given detector, but there is no model selection considered in SRS-based methods. **METAOD** is a model selection method using meta-learning without considering error rate control, while the **AutoMS** can do both model selection and error rate control, as shown in Table 1. For more details on the baseline methods, Refer to **Appendix B**.

## 5.2 Evaluation of the Jackknife

Firstly, We would like to show the advantage of Jackknife prediction with a toy example using synthetic data. We generate a training dataset $\mathcal{D}$ and a test dataset $\mathcal{U}$ from a Gaussian mixture model, where their sample sizes are $m = n = 2000$ respectively and $\mathcal{U}$ contains $10\%$ outliers. More details of the setting can be found in Sect. 5.3. In Fig. 1, we compare the Jackknife prediction to the single-random-splitting (SRS) method in Bates et al. [17] with different split ratios when the detector $\mathcal{M}$ is the OCSVM with its default hyperparameter in the PyOD package [28].

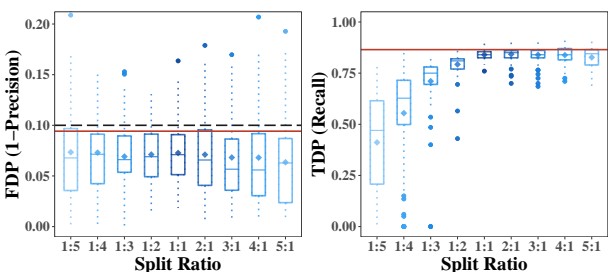

Figure 1: Performance of SRS-OCSVM [17] under different split ratios. Each box-plot shows the distribution of FDP and TDP under 100 simulations. The red line shows the corresponding results from Jackknife prediction. The black dashed line is the target FDR level $\alpha = 0.1$.

Focusing on one training set $\mathcal{D}$, we randomly split $\mathcal{D}$ 100 times for each split ratio of SRS method while Jackknife technique only has one prediction result. We can see from Fig. 1 that the Jackknife provides a quite tight FDP control and achieves a high TDP while finding more true outliers. As discussed above, the SRS-based methods depend on the random split and the split ratio and are usually unstable due to the randomness of the segmentation, making it hard to reproduce.

The Jackknife estimate helps improve the stability and the accuracy of the scores due to the usage of more samples to train the score function. Meanwhile, the Jackknife procedure will require a bit more computation compared to one-time data splitting, if $m$ is not too small. As the Jackknife procedure for $S_{\mathcal{M}}^{[-j]}(\cdot)$ is trained offline, one can always replace the Jackknife with cross-validation to reduce the computing burden to a certain extent. Instead of omitting one observation, we can randomly partition $\mathcal{D}$ into $k$ equal-sized folds and remove one fold from $\mathcal{D}$ as the subsample to train the score function. In the following of this section, we use $k = 100$ folds in the AutoMS-JK procedure.

## 5.3 Synthetic Data

**Data Description.** We consider the same synthetic setting as in Bates et al. [17]. We will compare the performance of different methods (the proposed AutoMS-JK, AutoMS-SRS, SRS-based methods and METAOD) when the target FDR level is set to $\alpha = 0.1$. Then we will show the results under different target FDR levels. The data are generated by sampling each data point $Z_i \in \mathbb{R}^{50}$ from a multivariate Gaussian mixture model, $Z_i = W_i + \sqrt{a}V_i$, where $W_i$ is one of $50$ cluster centers sampled

independently from the uniform distribution on $[-3, 3]^{50}$ and $V_i \in \mathbb{R}^{50}$ are i.i.d from $N(0, 1)$. The difference between inliers and outliers is the constant $a$ with $\mathbb{H}_0 : a = 1$ v.s. $\mathbb{H}_1 : a = 2$. We generate a clean dataset $\mathcal{D}$ with $m = 2000$ observations and a test dataset $\mathcal{U}$ with $n = 2000$ observations in which 90% are sampled as inliers. We aim to identify the remaining 10% of outliers in $\mathcal{U}$.

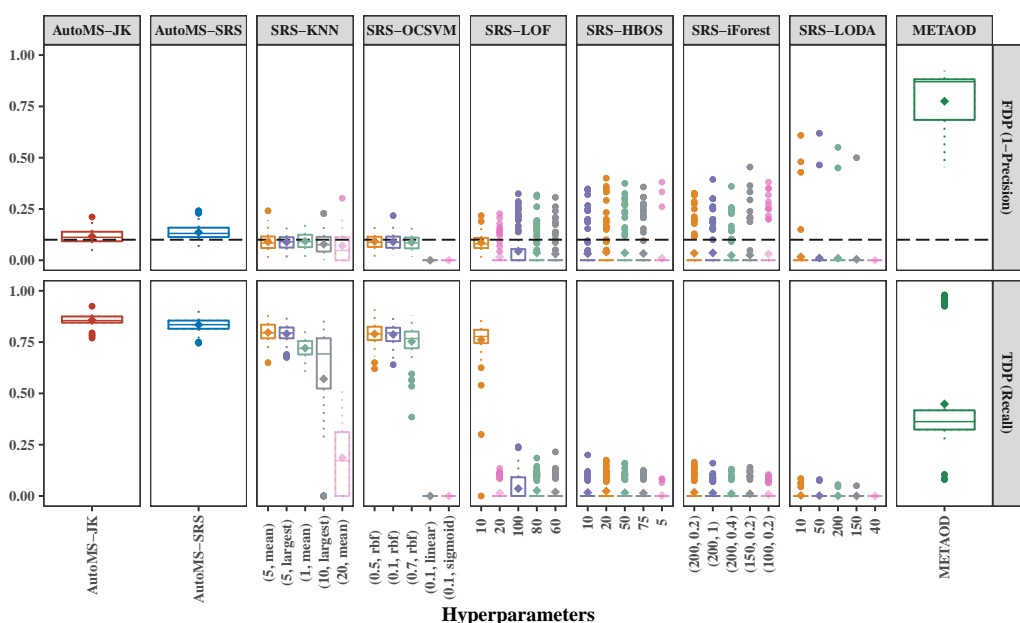

Figure 2: Comparisons of FDP (more stable around the dashed line) and TDP (the larger the better) among AutoMS-JK, AutoMS-SRS, SRS-based methods (with different hyperparameters) and METAOD. The dashed line is the target FDR level $\alpha$ which is set to $\alpha = 0.1$.

**Results.** With the given target FDR level $\alpha = 0.1$, the AutoMS (either AutoMS-JK or AutoMS-SRS) outperforms all baseline methods as shown in Fig. 2. (a) All the SRS-based methods can perfectly control the FDR with FDPs scattering around the given target FDR level $\alpha$. But in terms of detection power, the TDP performance of SRS-based methods vary greatly with different hyperparameters and different algorithms. Some SRS-based methods, such as SRS-LODA, SRS-HOBOS, SRS-iForest, even lose detection power with almost zero TDP/recall. This necessitates model selection among numerous novelty detection methods. (b) On the other hand, METAOD gives large FDPs, which means that most of the reported points are not true outliers. This phenomenon can be explained as METAOD does not employ any FDR control procedure. And the TDPs of METAOD do not perform well because the model selection of METAOD is based on historical data which may not be suitable for the data we use. (c) AutoMS-JK and AutoMS-SRS are superior to all SRS-based methods and METAOD with respect to the average TDR, with the average FDR slightly above the target FDR level $\alpha$, as it is more difficult to control the FDR of the selected detector than given detector. Additionally, AutoMS-JK gives a higher TDR than AutoMS-SRS, which indicates that using the Jackknife method instead of SRS enhances detection power. Although both AutoMS and SRS-based methods can control the FDR, AutoMS has better detection power with a larger TDR compared to SRS-based methods. (d) The reason why the FDR control of AutoMS is more difficult than SRS-based methods is that the FDP distribution of our selected detector is different from a given detector. Take a simple example: there are 10 standard Gaussian variables, each with a mean 0. The largest variable among those 10 does not have a mean 0. Therefore, the selected detector is more difficult to control FDR.

Fig. 3 shows the performance of AutoMS-JK, AutoMS-SRS and SRS-based-best methods under different target FDR levels. In particular, SRS-LOF-best is the best SRS-LOF detector with the largest TDR among all its candidate hyperparameters. Although all SRS-based-best methods can control the FDR under any target FDR level, the average TDPs of AutoMS-JK/AutoMS-SRS are higher than all the SRS-based-best methods at every target FDR level. The average FDP of AutoMS-JK/AutoMS-SRS is slightly larger than the given target FDR level $\alpha$, but their difference FDP $- \alpha$ does not increase as $\alpha$ becomes larger. The FDR of AutoMS-JK has a smaller deviation above the target $\alpha$ than AutoMS-SRS. Actually, in novelty detection tasks, it is not necessary to obtain procedures with

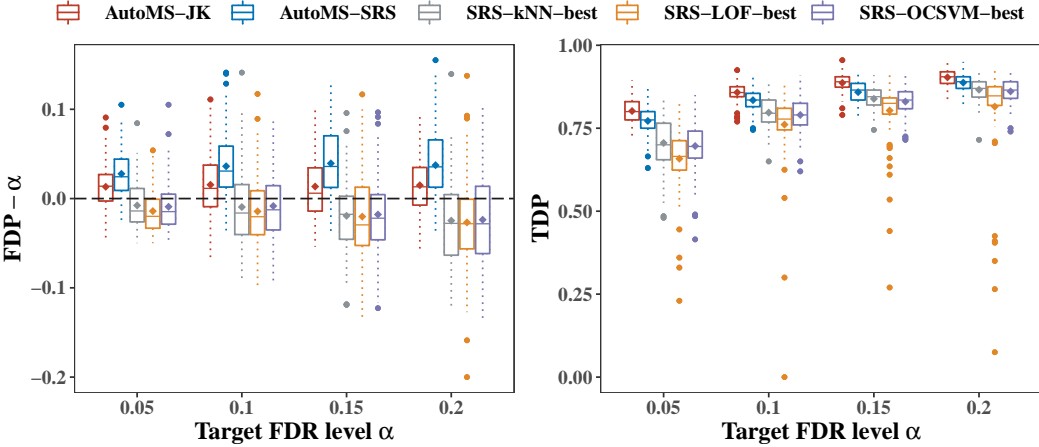

Figure 3: Performance of AutoMS-JK, AutoMS-SRS and SRS-based-best methods with different target FDR level $\alpha$. The difference between the FDP and the target FDR level $\alpha$ shows the deviation of FDP from $\alpha$.

finite FDR control, instead, a powerful and stable procedure that controls FDR asymptotically around the target FDR level $\alpha$ is good enough for most cases. That is what AutoMS does. AutoMS can uniformly offer much higher TDPs than SRS-based-best methods under different target FDR levels.

## 5.4 Real Data

We compare the proposed AutoMS with two different types of baselines: (i) model selection methods (METAOD [11]) and (ii) error-rate-controlled methods (SRS-based methods [17]), as in Table 1.

**Datasets.** Four real-world datasets for outlier detection are considered, and the details of the datasets are summarized in Table 2. We use empirical TDR (Recall) and empirical FDR (1-Precision) as performance measures. For real data implementation, we construct the training dataset $\mathcal{D}$ by randomly sampling $m = \min\{20000, N_0/3\}$ inlier samples from the total $N_0$ inliers. Then we focus on a test dataset generating mechanism under which $90\%$ of the $n = \min\{2000, (N_0 + N_1)/6\}$ observations in each test dataset $\mathcal{U}$ are sampled from $N_0$ inliers, and we seek to identify the remaining $10\%$ of outliers (i.e., $10\%$ outlier ratio). For results on 11 more real-world datasets [29] as used in Zhao et al. [11], refer to **Appendix D**.

Table 2: Summary of real-world datasets for novelty detection.

|  | Credit Card [30] | Covertype [31] | Satellite [32] | Shuttle [33] |
|---|---|---|---|---|
| # Features $d$ | 30 | 10 | 36 | 9 |
| # Inliers $N_0$ | 284,807 | 283,301 | 5,732 | 45,586 |
| # Outliers $N_1$ | 492 | 2747 | 703 | 12,414 |

**Results.** Fig. 4 shows the performance of different methods on real-world datasets, and Table 3 gives the empirical FDR and empirical TDR of different methods, where SRS-best is the best SRS detector with the largest TDR among all detectors. From Fig. 4 and Table 3, we can observe that AutoMS can maintain the FDPs well around the target FDR level $\alpha = 0.1$, while METAOD has inflated the empirical FDP to values around or above $0.5$ on all real-world datasets, indicating that in the detected set $\widehat{\mathcal{O}}$ obtained by METAOD, around $50\%$ of the detected outliers are not true outliers. It is worth mentioning that, all SRS-based methods can maintain the FDR level around $0.1$ as well as expected from their theoretical guarantees. As for true detection rate (TDR), AutoMS has reasonable empirical TDR on all datasets, while SRS-based methods sometimes perform too conservatively, have no detected outliers and their performance varies greatly from dataset to dataset.

In addition, the performances of SRS-LODA-best and SRS-LOF-best are rather different, particularly on the "Covertype" dataset and "Credit Card" dataset, which means SRS-based-best methods can not give a consistent good result on different tasks and its performance can be severely affected by the base detector the SRS used. On the other hand, AutoMS keeps accurate and stable FDR levels on all datasets and has reasonable TDR for different tasks. Overall, AutoMS can uniformly offer much higher TDRs than SRS-based-best methods across all settings and AutoMS can maintain the FDR

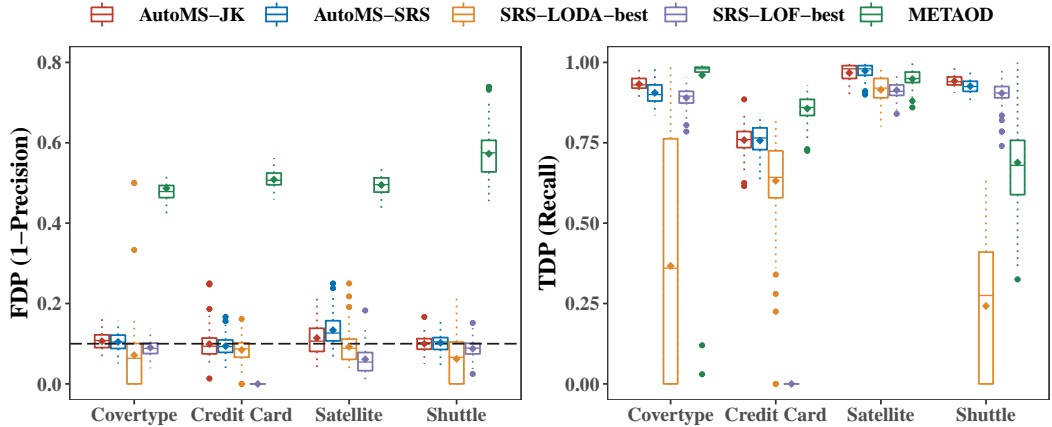

Figure 4: Performance on four different real-world datasets. Each box-plot shows the distribution of FDP and TDP. The dashed line is the target FDR level $\alpha = 0.1$.

level with a comparable TDR compared with METAOD. Table 4 shows results under different target FDR $\alpha$ levels (refer to **Appendix C** for more results and details).

Table 3: Empirical FDR and TDR of different methods when the target FDR level is set $\alpha = 0.1$.

| Dataset | AutoMS-JK | | AutoMS-SRS | | SRS-best | | METAOD | |
|---|---|---|---|---|---|---|---|---|
| | $\widehat{\text{FDR}}$ | $\widehat{\text{TDR}}$ | $\widehat{\text{FDR}}$ | $\widehat{\text{TDR}}$ | $\widehat{\text{FDR}}$ | $\widehat{\text{TDR}}$ | $\widehat{\text{FDR}}$ | $\widehat{\text{TDR}}$ |
| Covertype | 0.106 | 0.933 | 0.105 | 0.905 | 0.090 | 0.890 | 0.487 | 0.960 |
| Credit Card | 0.099 | 0.758 | 0.094 | 0.757 | 0.086 | 0.674 | 0.508 | 0.856 |
| Satellite | 0.114 | 0.968 | 0.134 | 0.974 | 0.069 | 0.963 | 0.495 | 0.948 |
| Shuttle | 0.100 | 0.942 | 0.102 | 0.926 | 0.088 | 0.914 | 0.573 | 0.689 |

Table 4: Empirical FDR and TDR under different target FDR levels $\alpha$ on "Shuttle" dataset.

| Shuttle | $\alpha = 0.05$ | | $\alpha = 0.10$ | | $\alpha = 0.15$ | | $\alpha = 0.20$ | |
|---|---|---|---|---|---|---|---|---|
| Method | $\widehat{\text{FDR}}$ | $\widehat{\text{TDR}}$ | $\widehat{\text{FDR}}$ | $\widehat{\text{TDR}}$ | $\widehat{\text{FDR}}$ | $\widehat{\text{TDR}}$ | $\widehat{\text{FDR}}$ | $\widehat{\text{TDR}}$ |
| AutoMS-JK | 0.042 | **0.883** | 0.100 | **0.942** | 0.151 | **0.951** | 0.209 | **0.953** |
| AutoMS-SRS | 0.051 | 0.659 | 0.102 | **0.926** | 0.159 | **0.952** | 0.216 | **0.956** |
| SRS-HBOS-best | 0.024 | 0.400 | 0.083 | 0.554 | 0.131 | 0.733 | 0.177 | 0.748 |
| SRS-OCSVM-best | 0.010 | 0.139 | 0.088 | 0.914 | 0.133 | 0.942 | 0.177 | 0.946 |
| METAOD | 0.573 | **0.689** | 0.573 | 0.689 | 0.573 | 0.689 | 0.573 | 0.689 |

# 6 Conclusion

The detector selection problem in novelty detection analysis is a challenging problem and it has not been thoroughly investigated in the literature, especially in terms of theoretical guarantees. In this paper, we propose an effective data-driven procedure, named AutoMS, that can select the "best" detector which offers high TDR and controls FDR simultaneously. The AutoMS is fully automatic without any supervision and is easily implemented. We give theoretical results showing that the proposed AutoMS can achieve the non-asymptotic bounds of FDP and maintain FDR control under mild conditions. Extensive experiments on synthetic and real-world datasets show AutoMS significantly improves the detection power and yields accurate FDR control.

## Acknowledgments and Funding Disclosure

We thank anonymous area chair and reviewers for their helpful comments. Changliang Zou's research was supported by the National Natural Science Foundation of China Grants (Nos. 11925106, 12231011, 11931001 and 11971247). Haojie Ren's research was supported by the National Natural Science Foundation of China Grants (No.12101398) and the Shanghai Sailing Program. This work was partially supported by the Technology Innovation Institute and MBZUAI joint project (NO. TII/ARRC/2073/2021): Energy-based Probing for Spiking Neural Networks.

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
