# Supplementary Material for "AutoMS: Automatic Model Selection for Novelty Detection with Error Rate Control"

Here we list all the terms we use in the paper.

- FDP, false discovery proportion (FDP), is the proportion of the normal samples that are erroneously detected as outliers.

$$\text{FDP} = \frac{|\widehat{\mathcal{O}}_{\mathcal{M}} \bigcap \mathcal{O}^c|}{1 \vee |\widehat{\mathcal{O}}_{\mathcal{M}}|},$$

where $\mathcal{O} \subset \mathcal{U}$ is the outlier set, $\mathcal{O}^c \subset \mathcal{U}$ is the inlier set, $\mathcal{U} = \mathcal{O} \cup \mathcal{O}^c$, $\widehat{\mathcal{O}}_{\mathcal{M}}$ is the decision set containing samples detected as outliers.

- TDP, true discovery proportion (TDP), is the proportion of the outliers that are correctly detected,

$$\text{TDP} = \frac{|\widehat{\mathcal{O}}_{\mathcal{M}} \bigcap \mathcal{O}|}{|\mathcal{O}|}.$$

- FDR, false discovery rate (FDR), is the expectation of FDP,

$$\text{FDR} = \mathbb{E}(\text{FDP}).$$

- TDR, true discovery rate (TDR), is the expectation of TDP,

$$\text{TDR} = \mathbb{E}(\text{TDP}).$$

- Precision, is the fraction of inliers that are correctly detected among the samples are detected. And FDP=1-Precision.
- Recall/Power, is the fraction of outliers that are correctly detected among the true outliers. And TDP= Recall = Power. Power = TDP.
- $\widehat{\text{FDR}}$ is the empirical FDR, which is the average FDP over 100 repetitions.
- $\widehat{\text{TDR}}$ is the empirical TDR, which is the average TDP over 100 repetitions.

## A  AutoMS Detector Set

We give an example of the detector set $\mathcal{G}$. We use 6 well-known novelty detection methods: Histogram-Based Outlier Score (HBOS), isolation Forest (iForest), K Nearest Neighbors (KNN), Lightweight On-line Detector of Anomalies (LODA), Local Outlier Factor (LOF) and One-Class Support Vector Machine (OCSVM), combined with their corresponding hyperparameters as a set of detectors $\mathcal{G}$ with 83 candidate models in our experiments. Those detectors are available in the Python "**PyOD**" [1] package. A complete list of the detector set $\mathcal{G}$ is shown in Table S1.

Note that the size of detector set $\mathcal{G}$ can be infinite, and our AutoMS procedure can still be used.

Table S1: An example of the detector set $\mathcal{G}$: detection methods with their hyperparameters (with default hyperparameters in bold).

| Method | Article | hyperparameters |
|--------|---------|-----------------|
| HBOS | Goldstein and Dengel [2] | n_histograms:{5,**10**,20,30,40,50,75,100} |
| iForest | Liu et al. [3] | n_estimators:{50,**100**,150,200} |
| | | max_feature:{0.2,0.4,0.6,0.8,**1.0**} |
| KNN | Ramaswamy et al. [4] | n_neighbors:{1,**5**,10,15,20,50,75,100} |
| | | method:{**"largest"**,"mean"} |
| LODA | Pevnỳ [5] | n_bins: {**10**,20,30,40,50,75,100,150,200} |
| LOF | Breunig et al. [6] | n_neighbors:{10,**20**,30,40,50,60,70,80,90,100} |
| OCSVM | Schölkopf et al. [7] | $\nu$:{0.1,0.3,**0.5**,0.7,0.9} |
| | | kernel:{**"rbf"**,"linear","poly","sigmoid"} |

# B   Experiment Setting with Details

To evaluate the performance of the proposed **AutoMS** method, we compare the proposed AutoMS with two different types of baselines: (i) model selection methods (METAOD [8]) and (ii) error-rate-controlled methods (SRS-based methods [9]), as in Table 1. (i) For model selection methods, we use METAOD [8] as a representative, since it deals with automatic unsupervised outlier model selection which uses the performances of historical outlier detection benchmark datasets as prior experience to automatically do model selection via meta-learning. (ii) For error-rate-controlled methods, we choose SRS-based methods [9] as typical methods to give statistical guarantees for the error rate control, but there is no model selection considered in SRS-based methods.

We consider two versions of the proposed AutoMS: **AutoMS-JK** and **AutoMS-SRS**, as shown in Table 1, where AutoMS-JK uses Jackknife technique and AutoMS-SRS replaces the Jackknife technique with single-random-splitting (SRS) technique. We explore the performances of the four methods with respect to FDR and TDR.

We give the detailed introduction for the four methods: (1) **SRS** [9], **METAOD** [8], (3) **AutoMS-JK** and (4) **AutoMS-SRS**.

(1) **SRS** [9] employs the single-random-splitting (SRS) based method in conjunction with a fixed detector, and constructs conformal p-values to control the FDR. We combine SRS with different detectors from the same $\mathcal{G}$ we use for AutoMS. In particular, SRS-LOF means that we use LOF as the base detector in combination with SRS, while SRS-LOF-best is the best SRS-LOF detector with the largest TDR among all its candidate hyperparameters. **SRS-based** methods [9] are representative error-rate-controlled methods which combine SRS with different base detectors from $\mathcal{G}$ as used for AutoMS. It is worth mentioning that SRS is guaranteed with FDR control for any given detector. But there is no model selection considered in SRS-based methods.

(2) **METAOD** [8] is a model selection algorithm which uses meta-learning to learn the performance of a large body of detection models on historical outlier detection benchmark datasets, and carries over this prior experience to automatically select an effective model to be employed on a new dataset. **METAOD** is a representative model selection method using meta-learning without considering error rate control, while the **AutoMS** can do both model selection and error rate control.

(3) **AutoMS-JK** is our proposed model selection procedure using Jackknife technique to select the "best" detector among a bunch of candidate detectors while controlling the FDR.

(4) **AutoMS-SRS** is a variant of AutoMS that replaces the Jackknife technique with single-random-splitting (SRS) to incorporate our model selection procedure.
As the single-random-splitting (SRS) can be combined with our model selection procedure, hereafter called AutoMS-SRS. AutoMS-SRS can be regarded as a special case of AutoMS and also has the theoretical guarantees that the selected model yields **asymptotically** valid FDR control. But SRS does not fully explore the clean data and can cause randomness by data-splitting. So we use the Jackknife method instead of SRS to improve the accuracy and stability of the estimated p-values and enhance detection power. Specially, when the sample size is too large, we use AutoMS-SRS to reduce the computation complexity.

# C   Detailed Experiment Results on 4 Real-world Datasets in the Main Paper

In this part, we provide detailed experiment results on the four real-world datasets, and we provide additional experiment results on 11 more real-world datasets in the next part.

We show the empirical FDR and TDR as performance measures of different methods on four real-world datasets with different target FDR levels $\alpha$ in Table S2, as a detailed extension of Table 3 and Table 4 in Sect. 5 of the main paper. As introduced in Sect. 5, the empirical FDR and TDR are calculated as the expectation of FDP and TDP, respectively. Note the performance of METAOD is same under different $\alpha$ since METAOD does not employ any FDR control procedure.

Table S2: Empirical FDR and TDR of AutoMS-JK, AutoMS-SRS, SRS-based methods and METAOD on four real-world datasets under different target FDR levels $\alpha$ (extension of Table 3 and Table 4).

| Method | $\alpha = 0.05$ | | $\alpha = 0.10$ | | $\alpha = 0.15$ | | $\alpha = 0.20$ | |
| --- | --- | --- | --- | --- | --- | --- | --- | --- |
| | $\widehat{\text{FDR}}$ | $\widehat{\text{TDR}}$ | $\widehat{\text{FDR}}$ | $\widehat{\text{TDR}}$ | $\widehat{\text{FDR}}$ | $\widehat{\text{TDR}}$ | $\widehat{\text{FDR}}$ | $\widehat{\text{TDR}}$ |
| **Covertype** | | | | | | | | |
| AutoMS-JK | 0.056 | **0.905** | 0.106 | **0.933** | 0.158 | **0.943** | 0.209 | **0.948** |
| AutoMS-SRS | 0.055 | 0.861 | 0.105 | 0.905 | 0.164 | 0.930 | 0.210 | 0.932 |
| SRS-HBOS-best | 0.000 | 0.000 | 0.062 | 0.001 | 0.138 | 0.007 | 0.221 | 0.011 |
| SRS-iForest-best | 0.033 | 0.011 | 0.090 | 0.022 | 0.102 | 0.045 | 0.165 | 0.042 |
| SRS-KNN-best | 0.046 | 0.802 | 0.092 | 0.871 | 0.135 | 0.902 | 0.181 | 0.922 |
| SRS-LODA-best | 0.012 | 0.146 | 0.072 | 0.367 | 0.116 | 0.593 | 0.165 | 0.759 |
| SRS-LOF-best | 0.046 | 0.848 | 0.090 | 0.890 | 0.135 | 0.912 | 0.177 | 0.926 |
| SRS-OCSVM-best | 0.000 | 0.000 | 0.000 | 0.000 | 0.000 | 0.000 | 0.000 | 0.000 |
| METAOD | 0.487 | **0.960** | 0.487 | **0.960** | 0.487 | **0.960** | 0.487 | **0.960** |
| **Credit Card** | | | | | | | | |
| AutoMS-JK | 0.047 | **0.608** | 0.099 | **0.758** | 0.158 | 0.805 | 0.207 | **0.830** |
| AutoMS-SRS | 0.050 | 0.595 | 0.094 | 0.757 | 0.146 | **0.807** | 0.198 | **0.830** |
| SRS-HBOS-best | 0.041 | 0.469 | 0.086 | 0.674 | 0.132 | 0.762 | 0.175 | 0.797 |
| SRS-iForest-best | 0.040 | 0.226 | 0.087 | 0.460 | 0.131 | 0.606 | 0.180 | 0.704 |
| SRS-KNN-best | 0.000 | 0.000 | 0.006 | 0.002 | 0.050 | 0.001 | 0.243 | 0.006 |
| SRS-LODA-best | 0.035 | 0.345 | 0.085 | 0.632 | 0.129 | 0.746 | 0.178 | 0.797 |
| SRS-LOF-best | 0.000 | 0.000 | 0.000 | 0.000 | 0.000 | 0.000 | 0.150 | 0.000 |
| SRS-OCSVM-best | 0.000 | 0.000 | 0.000 | 0.000 | 0.000 | 0.000 | 0.000 | 0.000 |
| METAOD | 0.508 | **0.856** | 0.508 | **0.856** | 0.508 | **0.856** | 0.508 | **0.856** |
| **Satellite** | | | | | | | | |
| AutoMS-JK | 0.067 | **0.951** | 0.114 | **0.968** | 0.165 | **0.969** | 0.218 | **0.970** |
| AutoMS-SRS | 0.068 | **0.968** | 0.134 | **0.974** | 0.183 | **0.975** | 0.236 | **0.974** |
| SRS-HBOS-best | 0.046 | 0.720 | 0.099 | 0.761 | 0.149 | 0.787 | 0.212 | 0.823 |
| SRS-iForest-best | 0.039 | 0.866 | 0.086 | 0.898 | 0.134 | 0.920 | 0.178 | 0.932 |
| SRS-KNN-best | 0.041 | 0.941 | 0.069 | 0.963 | 0.105 | 0.969 | 0.152 | 0.971 |
| SRS-LODA-best | 0.050 | 0.876 | 0.092 | 0.915 | 0.134 | 0.935 | 0.182 | 0.950 |
| SRS-LOF-best | 0.024 | 0.883 | 0.061 | 0.913 | 0.106 | 0.931 | 0.164 | 0.947 |
| SRS-OCSVM-best | 0.000 | 0.000 | 0.000 | 0.000 | 0.000 | 0.000 | 0.000 | 0.000 |
| METAOD | 0.495 | 0.948 | 0.495 | 0.948 | 0.495 | 0.948 | 0.495 | 0.948 |
| **Shuttle** | | | | | | | | |
| AutoMS-JK | 0.042 | **0.883** | 0.100 | **0.942** | 0.151 | **0.951** | 0.209 | **0.953** |
| AutoMS-SRS | 0.051 | 0.659 | 0.102 | **0.926** | 0.159 | **0.952** | 0.216 | **0.956** |
| SRS-HBOS-best | 0.024 | 0.400 | 0.083 | 0.554 | 0.131 | 0.733 | 0.177 | 0.748 |
| SRS-iForest-best | 0.044 | 0.417 | 0.092 | 0.469 | 0.138 | 0.504 | 0.178 | 0.528 |
| SRS-KNN-best | 0.015 | 0.195 | 0.086 | 0.906 | 0.130 | 0.942 | 0.171 | 0.944 |
| SRS-LODA-best | 0.022 | 0.102 | 0.062 | 0.242 | 0.117 | 0.363 | 0.169 | 0.498 |
| SRS-LOF-best | 0.010 | 0.124 | 0.088 | 0.904 | 0.137 | 0.933 | 0.184 | 0.944 |
| SRS-OCSVM-best | 0.010 | 0.139 | 0.088 | 0.914 | 0.133 | 0.942 | 0.177 | 0.946 |
| METAOD | 0.573 | **0.689** | 0.573 | 0.689 | 0.573 | 0.689 | 0.573 | 0.689 |

From Table S2, we see that the empirical FDR values of AutoMS (either AutoMS-JK or AutoMS-SRS) and SRS-based methods are all around the target FDR levels for all datasets. In finite samples, the empirical FDR of AutoMS is slightly larger than the given target FDR level $\alpha$, but their difference does not increase as $\alpha$ becomes larger. We also list the Standard Deviation for FDP and TDP of AutoMS-JK, AutoMS-SRS, SRS-based methods and METAOD on four real-world datasets under different target FDR levels $\alpha$ in Table S3.

Moreover, with similar target FDR level $\alpha$, AutoMS can uniformly offer much higher TDRs than SRS-based methods across all settings, as the former automatically selects the most suitable model

Table S3: Standard deviation for empirical FDR and TDR of AutoMS-JK, AutoMS-SRS, SRS-based methods and METAOD on four real-world datasets under different target FDR levels $\alpha$ .

| Method | $\alpha = 0.05$ | | $\alpha = 0.10$ | | $\alpha = 0.15$ | | $\alpha = 0.20$ | |
| --- | --- | --- | --- | --- | --- | --- | --- | --- |
| | $\widehat{\text{FDR}}$ | $\widehat{\text{TDR}}$ | $\widehat{\text{FDR}}$ | $\widehat{\text{TDR}}$ | $\widehat{\text{FDR}}$ | $\widehat{\text{TDR}}$ | $\widehat{\text{FDR}}$ | $\widehat{\text{TDR}}$ |
| **Covertype** | | | | | | | | |
| AutoMS-JK | 0.017 | 0.022 | 0.022 | 0.021 | 0.026 | 0.021 | 0.027 | 0.019 |
| AutoMS-SRS | 0.018 | 0.036 | 0.023 | 0.034 | 0.026 | 0.029 | 0.027 | 0.020 |
| SRS-HBOS-best | 0.000 | 0.000 | **0.228** | 0.003 | **0.285** | 0.016 | **0.365** | 0.022 |
| SRS-iForest-best | 0.075 | 0.019 | 0.137 | 0.030 | 0.131 | 0.051 | 0.233 | 0.048 |
| SRS-KNN-best | 0.018 | 0.045 | 0.024 | 0.033 | 0.028 | 0.022 | 0.036 | 0.021 |
| SRS-LODA-best | 0.024 | 0.269 | 0.127 | **0.380** | 0.109 | **0.366** | 0.062 | **0.279** |
| SRS-LOF-best | 0.019 | 0.042 | 0.021 | 0.034 | 0.027 | 0.025 | 0.029 | 0.024 |
| SRS-OCSVM-best | 0.000 | 0.000 | 0.000 | 0.000 | 0.000 | 0.000 | 0.000 | 0.000 |
| METAOD | **0.066** | **0.128** | 0.066 | 0.128 | 0.066 | 0.128 | 0.066 | 0.128 |
| **Credit Card** | | | | | | | | |
| AutoMS-JK | 0.021 | 0.076 | **0.036** | 0.046 | 0.118 | 0.037 | 0.181 | 0.053 |
| AutoMS-SRS | 0.021 | 0.089 | 0.026 | 0.049 | 0.031 | 0.028 | 0.034 | 0.024 |
| SRS-HBOS-best | 0.020 | 0.114 | 0.028 | 0.071 | 0.029 | 0.049 | 0.032 | 0.032 |
| SRS-iForest-best | **0.030** | 0.114 | 0.029 | 0.095 | 0.035 | **0.081** | 0.039 | **0.063** |
| SRS-KNN-best | 0.000 | 0.000 | 0.028 | 0.008 | **0.224** | 0.004 | **0.371** | 0.012 |
| SRS-LODA-best | 0.028 | **0.240** | 0.029 | **0.145** | 0.035 | 0.066 | 0.036 | 0.046 |
| SRS-LOF-best | 0.000 | 0.000 | 0.000 | 0.000 | 0.000 | 0.000 | 0.366 | 0.000 |
| SRS-OCSVM-best | 0.000 | 0.000 | 0.000 | 0.000 | 0.000 | 0.000 | 0.000 | 0.000 |
| METAOD | 0.021 | 0.039 | 0.021 | 0.039 | 0.021 | 0.039 | 0.021 | 0.039 |
| **Satellite** | | | | | | | | |
| AutoMS-JK | **0.039** | 0.031 | 0.043 | 0.027 | 0.053 | 0.029 | 0.043 | 0.027 |
| AutoMS-SRS | 0.030 | 0.024 | 0.040 | 0.024 | 0.045 | 0.023 | 0.039 | 0.023 |
| SRS-HBOS-best | 0.032 | **0.056** | **0.045** | **0.045** | **0.062** | **0.049** | **0.055** | **0.040** |
| SRS-iForest-best | 0.022 | 0.033 | 0.040 | 0.036 | 0.047 | 0.026 | 0.052 | 0.024 |
| SRS-KNN-best | 0.018 | 0.026 | 0.031 | 0.012 | 0.039 | 0.008 | 0.049 | 0.007 |
| SRS-LODA-best | 0.029 | 0.054 | 0.042 | 0.038 | 0.049 | 0.033 | 0.046 | 0.026 |
| SRS-LOF-best | 0.016 | 0.035 | 0.034 | 0.026 | 0.042 | 0.026 | 0.042 | 0.024 |
| SRS-OCSVM-best | 0.000 | 0.000 | 0.000 | 0.000 | 0.000 | 0.000 | 0.000 | 0.000 |
| METAOD | 0.022 | 0.027 | 0.022 | 0.027 | 0.022 | 0.027 | 0.022 | 0.027 |
| **Shuttle** | | | | | | | | |
| AutoMS-JK | 0.013 | 0.030 | 0.022 | 0.017 | 0.027 | 0.019 | 0.030 | 0.020 |
| AutoMS-SRS | 0.021 | 0.149 | 0.021 | 0.024 | 0.024 | 0.024 | 0.027 | 0.028 |
| SRS-HBOS-best | 0.026 | 0.162 | 0.037 | **0.260** | 0.031 | **0.203** | 0.033 | **0.204** |
| SRS-iForest-best | 0.023 | 0.054 | 0.030 | 0.050 | 0.038 | 0.046 | 0.045 | 0.053 |
| SRS-KNN-best | 0.026 | **0.328** | 0.021 | 0.026 | 0.023 | 0.017 | 0.027 | 0.016 |
| SRS-LODA-best | 0.037 | 0.145 | 0.054 | 0.201 | 0.065 | 0.197 | 0.054 | 0.138 |
| SRS-LOF-best | 0.023 | 0.267 | 0.024 | 0.036 | 0.028 | 0.023 | 0.035 | 0.019 |
| SRS-OCSVM-best | 0.022 | 0.308 | 0.021 | 0.027 | 0.027 | 0.017 | 0.028 | 0.017 |
| METAOD | **0.067** | 0.169 | **0.067** | 0.169 | **0.067** | 0.169 | **0.067** | 0.169 |

while maintaining the FDR control. On the Satellite dataset, the SRS-KNN-best method achieves the same high empirical TDP as AutoMS. It is because the optimal algorithm and hyperparameter selected by AutoMS are almost the same as those used by SRS-KNN-best.

On the other hand, the METAOD algorithm has a higher empirical TDR (i.e. higher recall) than AutoMS in some settings (such as "Covertype" dataset and "Credit Card" dataset), while the empirical FDR (i.e. 1 - Precision) of METAOD can be as large as $50\%$, indicating that around $50\%$ of outliers detected by METAOD are not true outliers. That is, the recall of METAOD can be larger than AutoMS in some applications, while its precision can only be maintained around or smaller than $50\%$, which

results from the fact that there is no error control in METAOD. In conclusion, AutoMS is a more stable novelty detection method containing model selection and FDR control.

We can see that the larger the target FDR level $\alpha$, the larger the empirical TDR, which means the more relaxation of the target FDR level, the higher improvement of empirical TDR, indicating the trade-off between precision and recall of detection methods. Adjusting the target FDR level leaves us the trade-off that with a larger target FDR level $\alpha$ the detection method can detect more outliers but produce more false discoveries.

So we can adjust our target FDR level $\alpha$ to obtain the satisfied FDR and TDR. However, the performance of METAOD is same under different target FDR levels $\alpha$ since METAOD does not employ any FDR control procedure. And AutoMS can uniformly offer much higher TDRs than SRS-based-best methods across all FDR levels. Therefore, our AutoMS is a practical model selection method, which takes into account FDR and TDR.

## D   Additional Experiment Results on More Real-world Datasets

As for the experiment scale of our AutoMS method, we are here trying to make it clear that why there is no need for our AutoMS method to go through hundreds of datasets. As MetaOD uses meta-learning, METAOD requires a large number of datasets as the historical benchmarks to measure the similarity between the test dataset and benchmark datasets. And the different test-bechmark-dataset-similarity will effect the results of METAOD, while our AutoMS approach has no special requirements for datasets. We have shown experiments on 4 datasets in the main paper, and the proposed AutoMS method works on all other datasets, of course not only on these 4 datasets shown in the main paper, which can be obtained from Theorem 1 (FDR control) in our paper. We have conducted additional experiments on 11 more real-world datasets. The conclusion still remains the same that AutoMS can always control FDR while METAOD cannot. To our best knowledge, is the first effort of model selection for novelty detection with theoretical guarantees in the view of FDR control. Our proposed AutoMS method can select the best model and simultaneously control the error rate of the best model.

Table S4: Summary of 11 more real-world datasets.

| Dataset | $d$ | $N_0$ | $N_1$ | Outlier Ratio ($p$) |
|---|---|---|---|---|
| abalone | 7 | 2096 | 2081 | 49.7% |
| comm.and.crime | 101 | 1001 | 993 | 49.7% |
| imgseg | 18 | 1320 | 990 | 42.9% |
| letter.rec | 16 | 10022 | 9978 | 49.9% |
| magic.gamma | 10 | 12332 | 6688 | 35.2% |
| opt.digits | 62 | 3357 | 2263 | 40.2% |
| pageb | 10 | 4913 | 560 | 10.2% |
| skin | 3 | 194198 | 50859 | 20.8% |
| spambase | 57 | 2788 | 1813 | 39.3% |
| synthetic | 10 | 10000 | 10000 | 50.0% |
| wave | 21 | 3343 | 1657 | 33.1% |

We provide additional experiment results on 11 more real-world datasets which are pre-processed data from Emmott et al. [10]. And the details of 11 real-world datasets are summarized in Table S4. The outlier ratio of the original dataset is calculated by $p = \frac{N_1}{N_0 + N_1}$. We construct the training dataset $\mathcal{D}$ by random sampling $m = \min\{20000, N_0/3\}$ from the total $N_0$ inliers. As for the test dataset $\mathcal{U}$, we need $n = \min\{2000, (N_0 + N_1)/6\}$ samples in total to generate the test dataset. We fix the outlier ratio as $p$, then randomly sample $n \times (1 - p)$ inliers from the remaining $N_0 - m$ inliers, and randomly sample $n \times p$ outliers from $N_1$ outliers. When constructing the test dataset $\mathcal{U}$, the outlier ratio for each dataset is the same as the original dataset as shown in Table S4.

Note that there are only inliers in the training dataset $\mathcal{D}$, and there are $p$ percent of outliers in the test dataset $\mathcal{U}$. For example, for the "abalone" dataset, there are $49.7\%$ outliers in its test dataset $\mathcal{U}$, and the goal of novelty detection is to detect the $49.7\%$ outliers, using only clean data $\mathcal{D}$. The goal of model selection for unsupervised outlier detection is to select the best outlier detection model. Our

goal here is to select the best model and simultaneously to control the error rate of the best model. We propose AutoMS method to achieve the goal of model selection for unsupervised outlier detection with error rate control.

Table S5: Empirical FDR and TDR of AutoMS-JK, AutoMS-SRS, SRS-best and METAOD on 11 more real-world datasets when the target FDR level $\alpha = 0.1$, where SRS-best is the best detector of SRS-based methods with the largest TDR among all detectors.

| Dataset | Outlier Ratio ($p$) | AutoMS-JK | | AutoMS-SRS | | SRS-best | | METAOD | |
|---|---|---|---|---|---|---|---|---|---|
| | | $\widehat{FDR}$ | $\widehat{TDR}$ | $\widehat{FDR}$ | $\widehat{TDR}$ | $\widehat{FDR}$ | $\widehat{TDR}$ | $\widehat{FDR}$ | $\widehat{TDR}$ |
| abalone | 49.7% | 0.077 | 0.277 | 0.096 | 0.335 | 0.062 | 0.284 | 0.192 | 0.436 |
| comm.and.crime | 49.7% | 0.064 | 0.540 | 0.091 | 0.607 | 0.056 | 0.497 | 0.197 | 0.427 |
| imgseg | 42.9% | 0.046 | 0.510 | 0.122 | 0.544 | 0.059 | 0.496 | 0.230 | 0.448 |
| letter.rec | 49.9% | 0.052 | 0.419 | 0.056 | 0.269 | 0.048 | 0.243 | 0.273 | 0.339 |
| magic.gamma | 35.2% | 0.071 | 0.352 | 0.083 | 0.354 | 0.065 | 0.337 | 0.244 | 0.581 |
| opt.digits | 40.2% | 0.065 | 0.767 | 0.096 | 0.743 | 0.059 | 0.718 | 0.219 | 0.592 |
| pageb | 10.2% | 0.138 | 0.374 | 0.182 | 0.477 | 0.110 | 0.353 | 0.546 | 0.720 |
| skin | 20.8% | 0.128 | 0.943 | 0.086 | 0.976 | 0.079 | 0.969 | 0.595 | 0.373 |
| spambase | 39.3% | 0.067 | 0.673 | 0.089 | 0.717 | 0.064 | 0.646 | 0.390 | 0.239 |
| synthetic | 50.0% | 0.059 | 0.994 | 0.064 | 0.994 | 0.050 | 0.991 | 0.225 | 0.460 |
| wave | 33.1% | 0.121 | 0.148 | 0.156 | 0.167 | 0.070 | 0.092 | 0.358 | 0.374 |

Table S5 shows the results of empirical FDR and empirical TDR of different methods (including AutoMS-JK, AutoMS-SRS, SRS-best and METAOD) on 11 more real-world datasets when the target FDR level $\alpha = 0.1$. As introduced above, these 11 real-world datasets have different outlier ratios which are shown in Table S4. The outlier ratio ranges from $10\%$ to $50\%$. Unsupervised novelty detection becomes more difficult when the outlier ratio goes higher, because only inliers are used in the cleaning training dataset $\mathcal{D}$ and more outliers need to be detected in the test dataset $\mathcal{U}$. The outlier ratio of the 11 real-world datasets ranges from $10\%$ to $50\%$. The experiment results in Table S5 show that at different outlier ratio, our AutoMS approach outperforms SRS-best and MATAOD for all the 11 real-world datasets.

Table S5 shows that the empirical TDR of AutoMS is higher than the empirical TDR of SRS-best since SRS-based methods only consider FDR control without model selection, while AutoMS selects the best detector with largest TDR while the empirical FDR is still controlled asymptotically around the FDR level.

The effectiveness of our AutoMS can be found and supported by the theories of our paper, which theoretically guarantees that AutoMS can control the FDR. It is worth noting that METAOD requires a large number of datasets as the historical benchmark to measure the similarity between the test dataset and benchmark datasets as it uses meta-learning.

All the empirical FDRs of all datasets using METAOD are as large as $0.2$ or even larger, indicating a very high false discovery rate, which means METAOD can not control the FDR. However, our proposed AutoMS can improve the TDR while controlling the FDR.

For the two versions of AutoMS, sometimes AutoMS-SRS has higher empirical TDR than AutoMS-JK, but the empirical FDR of AutoMS-SRS has a larger deviation above target FDR level $\alpha$ than AutoMS-JK. Table S6 gives the standard deviation of empirical FDR and empirical TDR of different methods on 11 more real-world datasets when the target FDR level $\alpha = 0.1$.

Table S6: Standard deviation of empirical FDR and empirical TDR with different methods (including AutoMS-JK, AutoMS-SRS, SRS-best and METAOD) on 11 more real-world datasets when the target FDR level $\alpha = 0.1$, where SRS-best is the best detector of SRS-based methods with the largest TDR among all detectors.

| Dataset | Outlier Ratio ($p$) | AutoMS-JK $\widehat{FDR}$ | $\widehat{TDR}$ | AutoMS-SRS $\widehat{FDR}$ | $\widehat{TDR}$ | SRS-best $\widehat{FDR}$ | $\widehat{TDR}$ | METAOD $\widehat{FDR}$ | $\widehat{TDR}$ |
|---|---|---|---|---|---|---|---|---|---|
| abalone | 49.7% | 0.040 | **0.110** | **0.050** | 0.087 | 0.033 | 0.090 | 0.031 | 0.051 |
| comm.and.crime | 49.7% | 0.026 | 0.072 | 0.029 | 0.055 | 0.031 | **0.158** | **0.044** | 0.083 |
| imgseg | 42.9% | 0.033 | 0.056 | **0.125** | **0.097** | 0.033 | 0.076 | 0.041 | 0.053 |
| letter.rec | 49.9% | 0.013 | 0.025 | 0.015 | 0.041 | 0.016 | **0.057** | **0.047** | 0.045 |
| magic.gamma | 35.2% | 0.020 | 0.026 | 0.023 | 0.028 | 0.021 | 0.032 | **0.029** | **0.064** |
| opt.digits | 40.2% | 0.023 | 0.039 | **0.117** | 0.067 | 0.020 | 0.099 | 0.075 | **0.122** |
| pageb | 10.2% | **0.086** | 0.100 | 0.077 | 0.095 | 0.068 | **0.118** | 0.039 | 0.071 |
| skin | 20.8% | 0.186 | 0.181 | 0.014 | 0.011 | 0.014 | 0.020 | **0.231** | **0.194** |
| spambase | 39.3% | 0.020 | 0.069 | **0.077** | 0.060 | 0.024 | **0.105** | 0.044 | 0.035 |
| synthetic | 50.0% | 0.007 | 0.003 | 0.008 | 0.003 | 0.008 | 0.002 | **0.016** | **0.015** |
| wave | 33.1% | **0.086** | 0.066 | 0.056 | 0.062 | 0.056 | 0.048 | 0.061 | **0.072** |

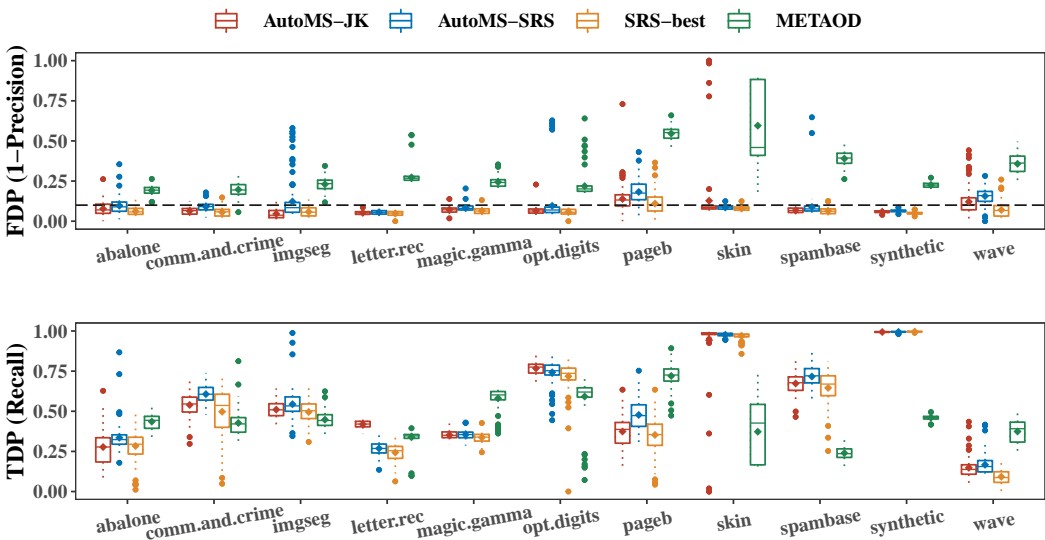

Figure S1: Performance on eleven different real-world datasets. Each box-plot shows the distribution of FDP and TDP. The dashed line is the target FDR level $\alpha = 0.1$.

Fig. S1 shows the performance of AutoMS-JK, AutoMS-SRS, SRS-best and METAOD on different real-world datasets when the target FDR level $\alpha = 0.1$. The empirical FDR of AutoMS-JK and AutoMS-SRS is all controlled around the given target FDR level $\alpha = 0.1$, The empirical FDR of AutoMS-JK has a smaller deviation above the target FDR $\alpha$ than AutoMS-SRS on "pageb" and "wave" datasets. The empirical TDR of AutoMS-JK and AutoMS-SRS is higher than the SRS-best methods at every outlier detections. Although the empirical TDR of METAOD is much higher than AutoMS, the empirical FDR is the worst as on error rate control has been taken into count. Therefore, our AutoMS procedure can be more practical in real-world applications, which takes into account FDR and TDR and selects the best detector.

# E Proofs

We present the proof of Our main theoretical result Theorem 1 on the validity of the AutoMS method for both FDP in the Appendix.

**Assumption 1** (Density). For any detector $\mathcal{M} \in \mathcal{G}$ and sufficiently large $m$, the conditional density of $S_{\mathcal{M}}(Z)$ given training dataset $\mathcal{D}$ is uniformly bounded above by a universal constant $c_f > 0$, where $Z \in \mathcal{U}$ is a new point.

**Assumption 2** (Learning stability). For any fixed $\mathcal{M}$, the score $S_{\mathcal{M}}$ satisfies: for large enough $m$,

$$\mathbb{P}(|S_{\mathcal{M}}(Z) - \tilde{\mu}_{\mathcal{M}}(Z)| \leq \kappa_m) \geq 1 - \zeta_m,$$

with some sequences satisfying $\kappa_m = o(1)$ and $\zeta_m = o(1)$ as $m \to \infty$, and some function $\tilde{\mu}_{\mathcal{M}}$.

**Assumption 3** (Rates). Denote $B_m = 2c_f\kappa_m + 3\zeta_m$. Suppose that $B_m = o(1)$, $n\varpi_m B_m = o(1)$ and $m\varpi_m / A_n^{2/3} = o(1)$.

**Theorem 1** (FDR control). *Suppose Assumptions 1–3 hold. Let $0 < \delta < 1$ and $0 < \alpha < 1$. There exist universal constants $C_1 > 0$ and $C_2 > 0$ so that the FDP of the proposed method satisfies*

$$\text{FDP}(L_{\mathcal{M}^*}) \leq \alpha \left[ 1 + \frac{4nB_m}{A_n} + C_1 \sqrt{\frac{\varpi_m}{\delta A_n^{2/3}}} + C_2 \sqrt{\frac{\varpi_m W_{mn}}{\delta}} \right], \tag{C.1}$$

*with probability at least $1 - 2\delta$, and*

$$\limsup_{(m,n)\to\infty} \text{FDR}(L_{\mathcal{M}^*}) \leq \alpha, \tag{C.2}$$

*where $W_{mn} = \left( \frac{n}{mA_n^{2/3}} + \frac{n}{A_n^{4/3}} \right) (1 + 15B_m + 2m^2 B_m)$.*

## E.1 Proof of Theorem

Before starting, we make clear the notations first. Let $\mathcal{D}$ and $\mathcal{U}$ be the training dataset and test dataset, respectively. Let $\mathcal{O}^c \subset \mathcal{U}$ be the subset of $\mathcal{U}$ containing all the inliers with size $n_0$, $\mathcal{O} \subset \mathcal{U}$ be the subset of all the outliers, and $\mathcal{O} \bigcup \mathcal{O}^c = \mathcal{U}$. When there is no risk of confusion, we also denote $\mathcal{U}$ the indices of the test dataset, $\mathcal{O}$ the indices of outliers, and $\mathcal{O}^c$ the indices of inliers. Let $S_{i,\mathcal{M}}$ be the short of $S_{\mathcal{M}}(Z_i)$ for $Z_i \in \mathcal{U}$ (i.e., $i \in \mathcal{U}$).

Note $S_{i,\mathcal{M}}$ and $S_{k,\mathcal{M}}$ are i.i.d. for each pair $(i, k) \in \mathcal{U}$ conditional on $\mathcal{D}$ and detector $\mathcal{M}$. We define $G_{\mathcal{M}}(t) = \mathbb{P}(S_{\mathcal{M}}(Z) > t \mid \mathcal{D})$, $G_{\mathcal{M}}^-(y) = \inf \{ t \geq 0 : G_{\mathcal{M}}(t) \leq y \}$ with $0 \leq y \leq 1$.

Define

$$G_{n,\mathcal{M}}(t) = n_0^{-1} \sum_{i\in\mathcal{O}^c} \mathbb{I}(S_{i,\mathcal{M}} > t),$$

where $n_0 = |\mathcal{O}^c|$ and recall

$$\mathcal{M}^* = \arg\max_{\mathcal{M}\in\mathcal{G}} |\widehat{O}_{\mathcal{M}}| = \arg\max_{\mathcal{M}\in\mathcal{G}} \sum_{i=1}^{n} \mathbb{I}(S_{i,\mathcal{M}} > L_{\mathcal{M}}),$$

$$L_{\mathcal{M}} = \inf \left\{ 0 < t \leq \bar{t}_{m,\mathcal{M}} : \frac{nG_{m,\mathcal{M}}(t)}{1 \vee \sum_{i=1}^{n} \mathbb{I}(S_{i,\mathcal{M}} > t)} \leq \alpha \right\}, \tag{C.3}$$

where $\bar{t}_{m,\mathcal{M}} = G_{m,\mathcal{M}}^-(A_n/n)$, $G_{m,\mathcal{M}}(t) = m^{-1}\sum_{j=1}^{m} \mathbb{I}(S_{j,\mathcal{M}}^{[-j]} > t)$ and $A_n = o(n)$ is a pre-specified sequence.

In the proof part, we use $c$ and $C$ denote strictly positive constants that can be changed from place to place, and they do not depend on $n$ and $m$. Let

$$\tilde{G}_{\mathcal{M}}(t) = \mathbb{E}\{G_{\mathcal{M}}(t)\}.$$

**Proof of Theorem 1.** Briefly, we first prove that the FDP of each detector can be bounded **uniformly** according to the uniform convergence properties Lemma C.1-C.3. Hence, the selected detector with

the largest number of discoveries will have the same non-asymptotic bounds for FDP. Then we give the asymptotic FDR control for the optimal detector.

By Lemma C.3-(ii) we have

$$\sup_{0 \le t \le \bar{t}_{\mathcal{M}}} \left| \frac{\tilde{G}_{\mathcal{M}}(t)}{G_{\mathcal{M}}(t)} - 1 \right| \le 4nB_m/A_n. \tag{C.4}$$

By Lemmas C.1 and C.2, we have

$$\sup_{0 \le t \le \bar{t}_{\mathcal{M}}} \left| \frac{G_{n,\mathcal{M}}(t)}{G_{\mathcal{M}}(t)} - 1 \right| \le C_1 \delta^{-1/2} \sqrt{\frac{\varpi_m}{A_n^{2/3}}}, \tag{C.5}$$

$$\sup_{0 \le t \le \bar{t}_{\mathcal{M}}} \left| \frac{G_{m,\mathcal{M}}(t)}{\tilde{G}_{\mathcal{M}}(t)} - 1 \right| \le C_2 \delta^{-1/2} \left\{ \varpi_m \left( \frac{n}{mA_n^{2/3}} + \frac{n}{A_n^{4/3}} \right) (1 + 15B_m + 2m^2 B_m) \right\}^{1/2} \tag{C.6}$$

hold simultaneously with probability at least $1 - 2\delta$.

Under the event that Eqs. (C.5) and (C.6) hold, we have

$$G_{m,\mathcal{M}}(\bar{t}_{\mathcal{M}}) \le (1 + g_{m,n})G_{\mathcal{M}}(\bar{t}_{\mathcal{M}}) \le (1 + g_{m,n})A_n/2,$$

where $g_{m,n} = o(1)$ under Assumption 3. Hence, with probability at least $1 - 2\delta$, for sufficiently large $m$ and $n$,

$$G_{m,\mathcal{M}}(\bar{t}_{\mathcal{M}}) \le G_{m,\mathcal{M}}(\bar{t}_{m,\mathcal{M}}) = A_n.$$

Since $G_{m,\mathcal{M}}(t)$ is a deceasing function of $t$, we have $\bar{t}_{m,\mathcal{M}} \le \bar{t}_{\mathcal{M}}$ with high probability, and hence $\bar{t}_{m,\mathcal{M}^*} \le \bar{t}_{\mathcal{M}^*}$. Similarly, $\bar{t}_{m,\mathcal{M}^*} \le \tilde{t}_{\mathcal{M}^*}$ with at least $1 - 2\delta$.

Write

$$\mathrm{FDP}(L_{\mathcal{M}^*}) = \frac{\sum_{i \in \mathcal{O}^c} \mathbb{I}(S_{i,\mathcal{M}^*} > L_{\mathcal{M}^*})}{1 \vee \sum_{i=1}^{n} \mathbb{I}(S_{i,\mathcal{M}^*} > L_{\mathcal{M}^*})}$$

$$= \frac{n_0}{n} \times \frac{nG_{m,\mathcal{M}^*}(L_{\mathcal{M}^*})}{1 \vee \sum_{i=1}^{n} \mathbb{I}(S_{i,\mathcal{M}^*} > L_{\mathcal{M}^*})} \times \frac{G_{\mathcal{M}^*}(L_{\mathcal{M}^*})}{\tilde{G}_{\mathcal{M}^*}(L_{\mathcal{M}^*})} \times \frac{G_{n,\mathcal{M}^*}(L_{\mathcal{M}^*})}{G_{\mathcal{M}^*}(L_{\mathcal{M}^*})} \times \frac{\tilde{G}_{\mathcal{M}^*}(L_{\mathcal{M}^*})}{G_{m,\mathcal{M}^*}(L_{\mathcal{M}^*})}$$

$$\le \alpha \times E_1 \times E_2 \times E_3 := \alpha R(L_{\mathcal{M}^*}).$$

In the case that $L_{\mathcal{M}^*} = \infty$, the result holds by definition. Thus, we only consider the case that $L_{\mathcal{M}^*} \le \bar{t}_{m,\mathcal{M}^*}$. Using Eqs.(C.4)-(C.6) to get bounds for the terms $E_1, E_2$ and $E_3$, respectively, Eq.(C.1) is proved.

Then, for any $\epsilon > 0$,

$$\mathrm{FDR} \le (1 + \epsilon)\alpha R(L_{\mathcal{M}^*}) + \mathbb{P}\left( \mathrm{FDP} \ge (1 + \epsilon)\alpha R(L_{\mathcal{M}^*}) \right),$$

which proves the second part of this theorem by using Assumption 3. $\qquad \square$

### E.2 Some Lemmas

We need the following lemmas to prove Theorem 1. The first lemma establishes the uniform convergence of $G_{n,\mathcal{M}}(t)$.

**Lemma C.1** (Uniform consistency of the estimator with $\mathcal{U}$). *Suppose Assumptions 1–3 hold. Denote $\bar{t}_{\mathcal{M}} = G_{\mathcal{M}}^{-}(A_n/(2n))$. For $\delta > 0$, we have with probability at least $1 - \delta$,*

$$\sup_{\mathcal{M} \in \mathcal{G}} \sup_{0 \le t \le \bar{t}_{\mathcal{M}}} \left| \frac{G_{n,\mathcal{M}}(t)}{G_{\mathcal{M}}(t)} - 1 \right| \le C \sqrt{\frac{\varpi_m}{\delta A_n^{2/3}}}.$$

*Proof.* For any given $\mathcal{M} \in \mathcal{G}$, note that $G_{\mathcal{M}}(t)$ is a deceasing and continuous function of $t$. Let $a_n = A_n/2$, $z_0 \le z_1 \cdots \le z_{h_n} \le 1$, and $t_k = G_{\mathcal{M}}^{-}(z_k)$, where $z_0 = a_n/n$, $z_k = a_n/n + b_n \exp(k^\tau)/n$, $h_n = \{\log((n - a_n)/b_n)\}^{1/\tau}$ with $b_n/a_n \to 0$ and $0 < \tau < 1$. Note that $|G_{\mathcal{M}}(t_k)/G_{\mathcal{M}}(t_{k+1}) - 1| \le Cb_n/a_n$ uniformly holds in $k$. It is therefore enough to obtain the convergence of

$$D_{n,\mathcal{M}} = \sup_{0 \le k \le h_n} \left| \frac{\sum_{i \in \mathcal{O}^c} \{\mathbb{I}(S_{i,\mathcal{M}} > t_k) - n_0 G_{\mathcal{M}}(t_k)\}}{n_0 G_{\mathcal{M}}(t_k)} \right|.$$

We calculate the second moment

$$
\begin{aligned}
D_{\mathcal{M}}(t) :&= \mathbb{E}\left(\left[\sum_{i \in \mathcal{O}^c}\{\mathbb{I}(S_{i,\mathcal{M}} > t) - G_{\mathcal{M}}(t)\}\right]^2 \mid \mathcal{D}\right) \\
&= \sum_{i \in \mathcal{O}^c}\mathbb{E}\left[\{\mathbb{I}(S_{i,\mathcal{M}} > t) - G_{\mathcal{M}}(t)\}^2 \mid \mathcal{D}\right] \\
&= n_0 G_{\mathcal{M}}(t)\{1 - G_{\mathcal{M}}(t)\} \\
&\leq n_0 G_{\mathcal{M}}(t).
\end{aligned}
\tag{C.7}
$$

From result (C.7) and Chebyshev's inequality, for $\epsilon > 0$ we have

$$
\begin{aligned}
\mathbb{P}(D_{n,\mathcal{M}} > \epsilon \mid \mathcal{D}) &\leq \sum_{k=0}^{h_n} \mathbb{P}\left(\left|\frac{\sum_{i \in \mathcal{O}^c}\{\mathbb{I}(S_{i,\mathcal{M}} > t_k) - G_{\mathcal{M}}(t_k)\}}{n_0 G_{\mathcal{M}}(t_k)}\right| \geq \epsilon \mid \mathcal{D}\right) \\
&\leq \frac{1}{\epsilon^2}\sum_{k=0}^{h_n}\frac{D_{\mathcal{M}}(t_k)}{\{n_0 G_{\mathcal{M}}(t_k)\}^2} \\
&\leq \frac{1}{\epsilon^2}\left\{\sum_{k=0}^{h_n}\frac{1}{n_0 G_{\mathcal{M}}(t_k)}\right\}.
\end{aligned}
$$

Moreover, observe that

$$
\sum_{k=0}^{h_n}\frac{1}{n_0 G_{\mathcal{M}}(t_k)} = \frac{n}{n_0}\left(\frac{1}{a_n} + \sum_{k=1}^{h_n}\frac{1}{a_n + b_n e^{k\tau}}\right) \leq Cb_n^{-1}.
$$

By Assumption 3,

$$
\mathbb{P}(\sup_{\mathcal{M} \in \mathcal{G}} D_{n,\mathcal{M}} > \epsilon) \leq \sum_{\mathcal{M} \in \mathcal{G}}\mathbb{P}(D_{n,\mathcal{M}} > \epsilon) \leq C\varpi_m/(\epsilon^2 b_n).
$$

Hence, conditional on $\mathcal{D}$, we have with probability at least $1 - \delta$,

$$
\begin{aligned}
\sup_{\mathcal{M} \in \mathcal{G}}\sup_{0 \leq t \leq \bar{t}_{\mathcal{M}}}\left|\frac{G_{n,\mathcal{M}}(t)}{G_{\mathcal{M}}(t)} - 1\right| &\leq C\max\left\{\sqrt{\varpi_m/(\delta b_n)}, b_n/a_n\right\} \\
&\leq C\sqrt{\frac{\varpi_m}{\delta a_n^{2/3}}},
\end{aligned}
$$

where we take $b_n = a_n^{2/3}$. Finally, note that this holds uniformly in $\mathcal{D}$, and thus we complete the proof. $\qquad\square$

**Lemma C.2** (Uniform consistency of the Jackknife estimator). *Suppose Assumptions 1–3 hold. Denote $\tilde{t}_{\mathcal{M}} = \tilde{G}_{\mathcal{M}}^-(A_n/2n)$. For $\delta > 0$, we have with probability at least $1 - \delta$,*

$$
\begin{aligned}
\sup_{\mathcal{M} \in \mathcal{G}}\sup_{0 \leq t \leq \tilde{t}_{\mathcal{M}}}&\left|\frac{\sum_i \mathbb{I}(S_{j,\mathcal{M}}^{[-j]} > t)}{m\tilde{G}_{\mathcal{M}}(t)} - 1\right| \\
&\leq C\delta^{-1}\left\{\varpi_m\left(\frac{n}{mA_n^{2/3}} + \frac{n}{A_n^{4/3}}\right)(1 + 15B_m + 2m^2 B_m)\right\}^{1/2}.
\end{aligned}
\tag{C.8}
$$

*Proof.* Define $G_{\mathcal{M}}^{[-j]}(t) = \mathbb{P}(S_{\mathcal{M}}^{[-j]}(Z) > t \mid \mathcal{D}^{[-j]})$, where $\mathcal{D}^{[-j]} = \mathcal{D} \setminus \tilde{X}_j$. Using the same partition technique in Lemma C.1, we need to obtain a bound for

$$
D_{m,\mathcal{M}} = \sup_{0 \leq k \leq h_n}\left|\frac{\sum_j\{\mathbb{I}(S_{j,\mathcal{M}}^{[-j]} > t_k) - \tilde{G}_{\mathcal{M}}(t_k)\}}{m\tilde{G}_{\mathcal{M}}(t_k)}\right|.
$$

Again, the main step is to bound

$$D_{\mathcal{M}}(t) = \mathbb{E}\left[\sum_j \left\{\mathbb{I}(S_{j,\mathcal{M}}^{[-j]} > t) - \tilde{G}_{\mathcal{M}}(t)\right\}\right]^2.$$

By the construction of Jackknife estimators on $\mathcal{D}$, $\{1 \le j \le m : S_{j,\mathcal{M}}^{[-j]}\}$ are not independent with each other. Observe for each pair $(j, k)$

$$
\begin{aligned}
&\mathbb{E}\left\{\mathbb{I}(S_{j,\mathcal{M}}^{[-j]} > t)\mathbb{I}(S_{k,\mathcal{M}}^{[-k]} > t)\right\} \\
&= \mathbb{E}\left[\mathbb{E}\left\{\mathbb{I}(S_{j,\mathcal{M}}^{[-j]} > t)\mathbb{I}(S_{k,\mathcal{M}}^{[-k]} > t) \mid \mathcal{D}^{[-j,-k]}\right\}\right] \\
&= \mathbb{E}\left\{\mathbb{P}\left(S_{j,\mathcal{M}}^{[-j]} > t, S_{k,\mathcal{M}}^{[-k]} > t \mid \mathcal{D}^{[-j,-k]}\right)\right\} \\
&\le \mathbb{E}\left\{\left|\mathbb{P}\left(S_{j,\mathcal{M}}^{[-j]} > t, S_{k,\mathcal{M}}^{[-k]} > t \mid \mathcal{D}^{[-j,-k]}\right) - \mathbb{P}\left(S_{j,\mathcal{M}}^{[-j,-k]} > t, S_{k,\mathcal{M}}^{[-j,-k]} > t \mid \mathcal{D}^{[-j,-k]}\right)\right|\right\} \\
&\quad + \mathbb{E}\left\{\mathbb{P}\left(S_{j,\mathcal{M}}^{[-j,-k]} > t, S_{k,\mathcal{M}}^{[-j,-k]} > t \mid \mathcal{D}^{[-j,-k]}\right)\right\} \\
&:= Q_1 + Q_2.
\end{aligned}
$$

By the fact that $S_{j,\mathcal{M}}^{[-j,-k]}$ and $S_{k,\mathcal{M}}^{[-j,-k]}$ are independent conditional on $\mathcal{D}^{[-j,-k]}$, we have $Q_2 = \mathbb{E}\{G_{\mathcal{M}}^{[-j,-k]}(t)\}^2$. In Lemma C.5, we set $Y_1 = S_{j,\mathcal{M}}^{[-j]}$, $Y_2 = S_{k,\mathcal{M}}^{[-k]}$, $Y_3 = S_{j,\mathcal{M}}^{[-j,-k]}$, and $Y_4 = S_{k,\mathcal{M}}^{[-j,-k]}$. Under Assumption 2, we have

$$Q_1 \le 4c_f\kappa_m + 6\zeta_m,$$

by applying Lemma C.5.

Accordingly,

$$
\begin{aligned}
D_{\mathcal{M}}(t) &:= \mathbb{E}\left[\sum_j \left\{\mathbb{I}(S_{j,\mathcal{M}}^{[-j]} > t) - G_{\mathcal{M}}^{[-j]}(t)\right\}\right]^2 + \mathbb{E}\left[\sum_j \left\{G_{\mathcal{M}}^{[-j]}(t) - \tilde{G}_{\mathcal{M}}(t)\right\}\right]^2 \\
&:= D_{1\mathcal{M}}(t) + D_{2\mathcal{M}}(t),
\end{aligned}
$$

where

$$
\begin{aligned}
D_{1\mathcal{M}}(t) &= \mathbb{E}\left[\sum_j G_{\mathcal{M}}^{[-j]}(t)\{1 - G_{\mathcal{M}}^{[-j]}(t)\}\right] \\
&\quad + \sum\sum_{j \ne k} \mathbb{E}\left[\left\{\mathbb{I}(S_{j,\mathcal{M}}^{[-j]} > t) - G_{\mathcal{M}}^{[-j]}(t)\right\}\left\{\mathbb{I}(S_{k,\mathcal{M}}^{[-k]} > t) - G_{\mathcal{M}}^{[-k]}(t)\right\}\right] \\
&\le \mathbb{E}\left\{\sum_j G_{\mathcal{M}}^{[-j]}(t)\right\} + \sum\sum_{j \ne k} \mathbb{E}\left\{\mathbb{I}(S_{j,\mathcal{M}}^{[-j]} > t)\mathbb{I}(S_{k,\mathcal{M}}^{[-k]} > t) - G_{\mathcal{M}}^{[-j]}(t)G_{\mathcal{M}}^{[-k]}(t)\right\} \\
&\le m\{\tilde{G}_{\mathcal{M}}(t) + B_m\} + \sum\sum_{j \ne k}\left[Q_1 + \mathbb{E}\{G_{\mathcal{M}}^{[-j,-k]}(t)\}^2 - \mathbb{E}\left\{G_{\mathcal{M}}^{[-j]}(t)G_{\mathcal{M}}^{[-k]}(t)\right\}\right] \\
&\le m\{\tilde{G}_{\mathcal{M}}(t) + B_m\} + m(m-1)\left\{2B_m + 2\tilde{G}_{\mathcal{M}}(t)B_m + 3B_m^2\right\},
\end{aligned}
$$

where we use Lemma C.3 to get

$$\left|\{G_{\mathcal{M}}^{[-j,-k]}(t)\}^2 - G_{\mathcal{M}}^{[-j]}(t)G_{\mathcal{M}}^{[-k]}(t)\right| \le 2\tilde{G}_{\mathcal{M}}(t)B_m + 3B_m^2.$$

Similarly, we have

$$D_{2\mathcal{M}}(t) = 9m^2 B_m^2.$$

With the above calculation and Chebyshev's inequality, for $\epsilon > 0$

$$\mathbb{P}(D_{m,\mathcal{M}} > \epsilon) \le \frac{1}{\epsilon^2} \sum_{k=0}^{h_n} \frac{D_{\mathcal{M}}(t_k)}{\{m\tilde{G}_{\mathcal{M}}(t_k)\}^2}$$

$$\le \frac{1}{\epsilon^2} \left\{ \sum_{k=0}^{h_n} \frac{1 + 2m^2 B_m}{m\tilde{G}_{\mathcal{M}}(t_k)} + \sum_{k=0}^{h_n} \frac{15 B_m}{\tilde{G}_{\mathcal{M}}^2(t_k)} \right\}.$$

Moreover, observe that

$$\sum_{k=0}^{h_n} \frac{1}{m\tilde{G}_{\mathcal{M}}(t_k)} \le Cn/(mb_n),$$

$$\sum_{k=0}^{h_n} \frac{1}{\tilde{G}_{\mathcal{M}}^2(t_k)} \le Cn/b_n^2,$$

which implies that

$$\mathbb{P}(D_{m,\mathcal{M}} > \epsilon) \le \frac{C}{\epsilon^2} \left( \frac{n}{mb_n} + \frac{n}{b_n^2} \right)(1 + 15 B_m + 2m^2 B_m).$$

By similar arguments in Lemmas C.1 and taking $b_n = a_n^{2/3}$, we can complete the proof. $\qquad\square$

**Lemma C.3.** *Suppose Assumption 1 and Assumption 2 hold. Then we have*

(i) *The following holds*

$$\mathbb{E}\left\{ \left| G_{\mathcal{M}}^{[-j]}(t) - G_{\mathcal{M}}(t) \right| \right\} \le 2c_f \kappa_m + 3\zeta_m.$$

*uniformly in $1 \le j \le m$.*

(ii) *It holds that $\mathbb{E}\left\{ \left| G_{\mathcal{M}}(t) - \tilde{G}_{\mathcal{M}}(t) \right| \right\} \le 4c_f \kappa_m + 6\zeta_m.$*

*Proof.* (i) Observe

$$\left| G_{\mathcal{M}}(t) - G_{\mathcal{M}}^{[-j]}(t) \right| = \left| \mathbb{P}(S_{\mathcal{M}}(Z) > t \mid \mathcal{D}) - \mathbb{P}(S_{\mathcal{M}}^{[-j]}(Z) > t \mid \mathcal{D}^{[-j]}) \right|$$

$$= \left| \mathbb{P}(S_{\mathcal{M}}(Z) > t \mid \mathcal{D}) - \mathbb{P}(S_{\mathcal{M}}^{[-i]}(Z) > t \mid \mathcal{D}) \right|.$$

The lemma follows immediately by using Assumption 2 and Lemma C.4 with $c_f > 0$.

(ii) By Lemma C.4, we have

$$|G_{\mathcal{M}}(t) - \tilde{G}_{\mathcal{M}}(t)| \le |G_{\mathcal{M}}(t) - \mathbb{P}(\tilde{\mu}_{\mathcal{M}}(Z) > t \mid \mathcal{D})| + |\mathbb{P}(\tilde{\mu}_{\mathcal{M}}(Z) > t) - \tilde{G}_{\mathcal{M}}(t)|$$

$$\le 4c_f \kappa_m + 3\zeta_m + 3\mathbb{P}(|S_{\mathcal{M}}(Z) - \tilde{\mu}_{\mathcal{M}}(Z)| > \kappa_m \mid \mathcal{D})$$

and the result holds by taking expectation and using Assumption 2 again. $\qquad\square$

**Lemma C.4** (Estimation difference). *Assume that two random variables $Y_1$ and $Y_2$ have density functions bounded by a constant $c > 0$, and satisfy $\mathbb{P}(|Y_1 - Y_2| \le \varepsilon) \ge 1 - \zeta$, where $\varepsilon > 0$ and $\zeta > 0$. Then for all $t > 0$, $|\mathbb{P}(Y_1 > t) - \mathbb{P}(Y_2 > t)| \le 2c\varepsilon + 3\zeta$.*

*Proof.* By the fact that

$$\mathbb{P}(Y_1 + \varepsilon > t, |Y_1 - Y_2| \le \varepsilon) \le \mathbb{P}(Y_2 > t, |Y_1 - Y_2| \le \varepsilon) \le \mathbb{P}(Y_1 - \varepsilon > t, |Y_1 - Y_2| \le \varepsilon),$$

we have

$$|\mathbb{P}(Y_1 > t) - \mathbb{P}(Y_2 > t)|$$
$$\le |\mathbb{P}(Y_1 > t) - \mathbb{P}(Y_2 > t, |Y_1 - Y_2| \le \varepsilon)| + \mathbb{P}(|Y_1 - Y_2| > \varepsilon)$$
$$\le |\mathbb{P}(Y_1 > t) - \mathbb{P}(Y_1 > t - \varepsilon, |Y_1 - Y_2| \le \varepsilon)| + |\mathbb{P}(Y_1 > t) - \mathbb{P}(Y_1 > t + \varepsilon, |Y_1 - Y_2| \le \varepsilon)|$$
$$\quad + \mathbb{P}(|Y_1 - Y_2| > \varepsilon)$$
$$\le |\mathbb{P}(Y_1 > t) - \mathbb{P}(Y_1 > t - \varepsilon)| + |\mathbb{P}(Y_1 > t) - \mathbb{P}(Y_1 > t + \varepsilon)| + 3\mathbb{P}(|Y_1 - Y_2| > \varepsilon)$$
$$\le 2c\varepsilon + 3\zeta,$$

where the last inequality comes from the Lipschitz condition. $\qquad\square$

**Lemma C.5.** *Assume that four random variables $Y_1$, $Y_2$, $Y_3$, and $Y_4$ have density functions bounded by a constant $c > 0$, and satisfy $\mathbb{P}(|Y_1 - Y_3| \le \varepsilon) \ge 1 - \zeta$ and $\mathbb{P}(|Y_2 - Y_4| \le \epsilon) \ge 1 - \varsigma$, where $\varepsilon > 0$, $\epsilon > 0$, $\zeta > 0$ and $\varsigma > 0$. Then for all $t > 0$,*

$$|\mathbb{P}(Y_1 > t, Y_2 > t) - \mathbb{P}(Y_3 > t, Y_4 > t)| \le 2c(\varepsilon + \epsilon) + 3(\zeta + \varsigma).$$

*Proof.* Define the events $\mathcal{E}_1 = \{|Y_1 - Y_3| \le \varepsilon\}$, $\mathcal{E}_2 = \{|Y_2 - Y_4| \le \epsilon\}$, and $\mathcal{E} = \mathcal{E}_1 \bigcup \mathcal{E}_2$. Note that $\mathbb{P}(Y_1 + \varepsilon > t, \mathcal{E}_1) \le \mathbb{P}(Y_3 > t, \mathcal{E}_1) \le \mathbb{P}(Y_1 - \varepsilon > t, \mathcal{E}_1)$ and $\mathbb{P}(Y_2 + \epsilon > t, \mathcal{E}_2) \le \mathbb{P}(Y_4 > t, \mathcal{E}_2) \le \mathbb{P}(Y_2 - \epsilon > t, \mathcal{E}_2)$. Then, we have

$$
\begin{aligned}
&\mathbb{P}(Y_1 > t, Y_2 > t) - \mathbb{P}(Y_3 > t, Y_4 > t) \\
&\le |\mathbb{P}(Y_1 > t, Y_2 > t) - \mathbb{P}(Y_3 > t, Y_4 > t, \mathcal{E})| + \mathbb{P}(\mathcal{E}_1^c) + \mathbb{P}(\mathcal{E}_2^c) \\
&\le |\mathbb{P}(Y_1 > t, Y_2 > t) - \mathbb{P}(Y_1 - \varepsilon > t, Y_2 - \epsilon > t, \mathcal{E})| \\
&\quad + |\mathbb{P}(Y_1 > t, Y_2 > t) - \mathbb{P}(Y_1 + \varepsilon > t, Y_2 + \epsilon > t, \mathcal{E})| + \mathbb{P}(\mathcal{E}_1^c) + \mathbb{P}(\mathcal{E}_2^c) \\
&\le |\mathbb{P}(Y_1 > t, Y_2 > t) - \mathbb{P}(Y_1 - \varepsilon > t, Y_2 - \epsilon > t)| \\
&\quad + |\mathbb{P}(Y_1 > t, Y_2 > t) - \mathbb{P}(Y_1 + \varepsilon > t, Y_2 + \epsilon > t| + 3\mathbb{P}(\mathcal{E}_1^c) + 3\mathbb{P}(\mathcal{E}_2^c) \\
&\le 2c(\varepsilon + \epsilon) + 3(\zeta + \varsigma),
\end{aligned}
$$

where we use the Lipschitz condition again. $\qquad\square$