# OpenReview forum: "AutoMS: Automatic Model Selection for Novelty Detection with Error Rate Control"
_NeurIPS.cc/2022/Conference — NeurIPS 2022 Accept_

### Official Review · Reviewer_VjGX · 2022-07-11

**Rating:** 6
**Confidence:** 2
**Soundness:** 2 fair
**Presentation:** 3 good
**Contribution:** 2 fair

**Summary:**

The authors propose a model selection method for novelty detection with false discovery rate (FDR) control. Given a detection model $M$, a detection threshold $L_M$ is selected based on the Benjamini-Hochberg (BH) procedure so that the FDR of $M$ is less than $\alpha$. To estimate the p-values in the BH procedure precisely, the authors propose to apply the Jackknife estimation, which extends the existing work by Bates et al. After estimating $L_M$ for each model $M \in G$, the model that most detects the novelties with $L_M$ is selected as the best model $M^*$. The authors also give theoretical results to show that the FDP of $M^*$ is non-asymptotically bounded and the FDR of $M^*$ is asymptotically bounded by $\alpha$. Experiments using synthetic or real datasets demonstrate the advantage of the proposed method against the work by Bates et al. or METAOD.


**Questions:**

It seems possible to apply the model selection using Equation (6) even when SRS is used to estimate $L_M$. If this is true, I would like to see the results of "model selection + SRS" in Fig.2-4.

In Fig.4:
- Why SRS-kNN and SRS-OCSVM are not compared?
- Does AutoMS include kNN/OCSVM models as candidates?

In Algorithm 1:
- Procedure of learning $S_M^{[-j]}$ should be added between line3-4.
- line 11: Input -> Output

In line 189, what is "the ADNER method"?


**Limitations:**

The computational overhead of applying the k-fold Jackknife against the original SRS should be assessed in the experiment.


**Strengths And Weaknesses:**

Strengths
- Hyperparameter tuning or model selection is especially hard in unsupervised settings like novelty dectecion. This paper proposes a simple yet effective approach for this problem from the viewpoint of "maximize_{M \in G} #detection(M), subject to FDR(M) \lt \alpha$."
- The control of FDR of $M$ is mainly achieved by the existing framework of Bates et al., but its Jackknife extention is proposed.

Weaknesses
- The computational overhead of applying the Jackknife procedure is not negligible, especially when the training set is large
- Experimental results, e.g. Fig.3, suggest that the FDR control of $M$ gets slightly worse by applying the Jackknife compared to the original SRS (by Bates et al.).


=====POST-REBUTTAL COMMENTS========

Thanks for the authors' response. The newly added experimental results and authors' response addressed a part of my concerns. I have raised my score.

---

> ### Author Response · Authors · 2022-08-02
> **Response to Reviewer VjGX**
>
> Q1: The computational overhead of applying the Jackknife procedure is not negligible, especially when the training set is large
>
> A1: We use Jackknife method to improve accuracy by making full use of data information, which inevitably makes some sacrifices in computational complexity. If computational considerations outweigh accuracy in practical applications, we also recommend using cross-validation instead of leave-one-out. Or we can combine our model selection procedure with SRS when computationally intensive.
>
> Q2: Experimental results, e.g. Fig.3, suggest that the FDR control of $\mathcal{M}$ gets slightly worse by applying the Jackknife compared to the original SRS (by Bates et al.).
>
> A2:
> - 1) The reason why the FDR control of AutoMS is more difficult than SRS is that the FDP distribution of our selected detector is different from a given detector. Take a simple example: there are 10 standard Gaussian variables, each with a mean 0. The largest variable among those 10 does not have a mean 0. Therefore, the selected detector is more difficult to control FDR.
>
> - 2) A smaller threshold could detect a larger number of discoveries including more false discoveries, which means the FDP and TDP change in the same direction. If we want a higher TDP, the FDP will tend to increase as well. From the Experiment results in **General Response**, both AutoMS-SRS and AutoMS-JK have this problem caused by the selection step. But we can see from Theorem 4.1 and the experimental results that the deviation of FDP is very small. Our procedure can select the detector that discovers more outliers while the FDP is still controlled **asymptotically around the FDR level**.
>
> - 3) SRS does not fully explore the clean data and can cause randomness by data-splitting. So we use the Jackknife method instead of SRS to improve the accuracy and stability of the estimated p-values and enhance detection power. On the other hand, SRS only considers FDR control while AutoMS selects the detector with largest TDR while the FDP is still controlled **asymptotically around the FDR level**. Therefore, our AutoMS strategy can be more practical in real applications, which takes into account FDR and TDR and selects the best detector.
>
> Q3: It seems possible to apply the model selection using Equation (6) even when SRS is used to estimate $L_{\mathcal{M}}$. If this is true, I would like to see the results of "model selection + SRS" in Fig.2-4.
>
> A3: SRS can be combined with our model selection procedure, hereafter called AutoMS-SRS. AutoMS-SRS can be regarded as a special case of AutoMS and also has the theoretical guarantees that the selected model yields asymptotically valid FDR control. Experiment results on AutoMS-SRS and AutoMS-JK is showed in **General Response**.
>
>
> Q4: In Fig.4: Why SRS-kNN and SRS-OCSVM are not compared?
>
> A4: We compared 6 algorithms under different target FDR levels $\alpha$ including kNN and OCSVM coupled with SRS in **Section B of Supplementary Material**, and we did not show the results of SRS-kNN and SRS-OCSVM because they sometimes give all zeros and they are not always giving usable results.
> We thus picked better-behaved SRS-LODA and SRS-LOF to compare with our method, and results show that the TDR of AutoMS is higher than the TDR of SRS-based methods.
>
>
> Q5: Does AutoMS include kNN/OCSVM models as candidates?
>
> A5: Yes, AutoMS use HBOS, iForest, kNN, LODA, LOF, and OCSVM with their corresponding hyperparameters as a set of candidate detectors. A complete list of the detector set we used is shown in **Section A of Supplementary Material**.
>
>
> Q6: In Algorithm 1: Procedure of learning $S_{\mathcal{M}}^{[-j]}$ should be added between line3-4.
>
> A6: The definition of $S_{\mathcal{M}}^{[-j]}$ is given in lines 139-140. Define $S_{\mathcal{M}}^{[-j]}$ as the score function trained on $\mathcal{D}^{[−j]}$ which is the subset of the training set $\mathcal{D}$ with the jth observation removed.

---

> > ### Comment · Reviewer_VjGX · 2022-08-08
> > **Thank you for your responses**
> >
> > Thank you for your responses. I understand that FDP of AutoMS is more difficult to control than that of SRS. Also, additional experiments suggest the advantage of applying the Jackknife procedure.
> > The newly added experimental results and authors' response addressed a part of my concerns. I have raised my score (4->6).

---

> > > ### Author Response · Authors · 2022-08-09
> > > **Thanks to reviewer VjGX for the thoughtful feedback**
> > >
> > > We greatly appreciate the time you spent on our responses and your valuable suggestions for adding experimental results. We will add experiments with AutoMS-SRS in the final version to further support the conclusion that our AutoMS-JK is better than SRS and AutoMS-SRS.

---

### Official Review · Reviewer_MLPQ · 2022-07-12

**Rating:** 4
**Confidence:** 4
**Soundness:** 3 good
**Presentation:** 3 good
**Contribution:** 2 fair

**Summary:**

This paper proposes a general AutoML framework for novelty detection and controlling the error rate of the model. The framework consists of an automated model selection procedure with FDR control. The theoretical bound is provided for AutoMS. Extensive experiments are conducted to demonstrate its effectiveness.

**Questions:**

None

**Strengths And Weaknesses:**

Strengths

1: The paper proposed a unified framework that can be combined with different base detectors
2: The paper provides a theoretical bound of FDR
3: Experiments are conducted to evaluate the effectiveness of AutoMS on both synthetic and real-world data.

Weaknesses
1. Only several real-world datasets are selected in the experiments. As a comparison, the previous work MetaOD has performed experiments on hundreds of datasets. The authors are encouraged to conduct a more thorough comparison with MetaOD.

---

> ### Author Response · Authors · 2022-08-02
> **Response to Reviewer MLPQ**
>
> Q: Only several real-world datasets are selected in the experiments. As a comparison, the previous work MetaOD has performed experiments on hundreds of datasets. The authors are encouraged to conduct a more thorough comparison with MetaOD.
>
> A:
> - 1) It is worth noting that METAOD is a model selection method using meta-learning without considering error rate control, while the AutoMS can do both model selection and error rate control. That is why METAOD requires a large number of datasets to study the effect of task similarity, while a few datasets are enough for our proposed AutoMS to illustrate the advantages.
> - 2) METAOD requires a large number of datasets as the historical benchmark to measure the similarity between the test set and benchmark datasets by using meta-learning. And the different train/test dataset similarity will effect the results of METAOD, while our AutoMS approach has no special requirements for datasets.
> - 3) The four real datasets used in section 5.4 have illustrated the advantages of AutoMS over other methods, therefore we donot need to go through hundreds of datasets.
> - 4) Note that SRS method is guaranteed with FDR control for any given detector, without considering model selection. The conclusion that our AutoMS approach outperforms SRS and MATAOD is unified for the four data.
> - 5) For example, Credit card is more suitable using SRS-LODA and Covertype is better using SRS-LOF, which reflects the importance of model selection. The FDP of all datasets using METAOD is very high, indicating a very high false discovery rate, which means METAOD can not control the FDR.
> However, AutoMS can improve the TDR while controlling the FDR.

---

> > ### Comment · Reviewer_MLPQ · 2022-08-08
> > **Thank you for the response**
> >
> > I thank the authors for the detailed response. I agree that AutoMS has the advantage of not requiring historical benchmarks. However, it is unclear to me whether AutoMS works only on these 4 datasets or it also works on other datasets. Additionally, since MetaOD is clearly an important (or maybe the only) previous work about model selection for OD, I believe a more thorough comparison with MetaOD on more datasets is needed to show the significance of the results. I keep my score unchanged since my concern is not addressed.

---

> > > ### Author Response · Authors · 2022-08-09
> > > **Response to Reviewer MLPQ's post-rebuttal comment**
> > >
> > > Thank you again for your concern on the experiments.
> > >
> > > 1. One advantage of AutoMS is not depending on historical benchmarks.
> > > - Clearly METAOD is an important previous work about model selection for OD.
> > > - As METAOD uses meta-learning, therefore, METAOD requires a large number of datasets as the historical benchmarks to measure the similarity between the test set and benchmark datasets.
> > > - It is quite obvious that our AutoMS has the advantage of not requiring historical benchmarks.
> > > - And the different train/test dataset similarity will effect the results of METAOD, while our AutoMS approach has no special requirements for datasets.
> > >
> > >
> > > 2. The most difference between our AutoMS method and METAOD is that AutoMS can control the FDR, which METAOD does not take into account FDR control.
> > > - The most important **advantage** of our AutoMS procedure is that **the AutoMS can do both FDR control and model selection while METAOD is a model selection method without considering error rate control**.
> > > - All datasets we use show that without FDR control, the points detected using METAOD may contain too many false discoveries, which already provides sufficient evidence for our conclusion.
> > > - We do not choose special datasets that show the FDR of AutoMS outperforming METAOD.
> > >
> > >
> > > 3. We think the experiments were sufficient enough to show the advantages of our procedure AutoMS compared to METAOD. We are willing to do more experiments if needed, but the experiments results are there to support our theory, not our main goal.
> > > - Our AutoMS method works on all other datasets, of course not only on these 4 datasets showed in the original paper. The effectiveness of our AutoMS can be found and supported by the theories of our paper, which AutoMS can control the FDR.
> > > - If needed, we can do experiments on as many datasets as we can in the final version. But the experiments are not our main goal in our AutoMS method, and they are used to support the effectiveness of AutoMS and our theories.
> > > - We believe the conclusion on more datasets remains that AutoMS can always control FDR while METAOD cannot.

---

> ### Author Response · Authors · 2022-08-09
> **Experiment scale of our AutoMS method and METAOD**
>
>
> Dear Reviewer MLPQ,
>
> Thank you for providing the insightful comments on the **Experiment scale of our AutoMS method and METAOD**.
> We have tried our best to answer your questions piece by piece, to make it clear that why there is no need for our AutoMS method to go through hundreds of datasets. As MetaOD uses meta-learning, therefore, METAOD requires a large number of datasets as the historical benchmarks to measure the similarity between the test set and benchmark datasets. We sincerely understand your concern on the effectiveness of our AutoMS method, as we have only showed experiments on 4 datasets. One thing is certain that our AutoMS method works on all other datasets, of course not only on these 4 datasets showed in the original paper, which can be obtained from Theorem 4.1 (FDR control) in our paper.
> We are willing to do more experiments if needed, but the conclusion will remain the same that AutoMS can always control FDR while METAOD cannot. We will revise the manuscript accordingly in the final version.
>
> We really appreciate it if you could re-evaluate our work which, to our best knowledge, is the first effort of model selection for novelty detection with theoretical guarantees in the view of FDR control. Our proposed AutoMS method can select the best model and simultaneously control the error rate of the best model.
>
>
> Authors of Paper AutoMS

---

### Official Review · Reviewer_47FE · 2022-07-18

**Rating:** 6
**Confidence:** 4
**Soundness:** 3 good
**Presentation:** 3 good
**Contribution:** 2 fair

**Summary:**

This work automatically selects a best detection model while simultaneously controlling the false discovery rate.
The experimental results shows that the proposed method can control the false discovery rate (FDR) and the true discovery rate (TDR) simultaneously.

**Questions:**

I have a couple of concerns and questions regarding the method, and empirical evaluations.

1. In page5 rationale,  you explain why we can select the model which detects the most outliers as the "best" one.  This idea seems heuristic somehow, but I am still confused about this part.  Consider there are two models  A and B,  and TDP_A =1 and TDP_B = 1. But B is more aggressive and have detect some false novelty.  Based on your strategy, we will choose B as the best,  but actually A is better than B because the FDP of A is smaller.  The issue may be  that you assume that all the models will keep the FDR level $\alpha$ constant   .

2. The main theoretical result is based on assumption 4.1-4.3. It is very common to require the assumption of learning stability and density. However, the assumption of rates seems  a little complicated here. Could you give more explanation about this assumption?

3. In section 5.2, the results shows that the Jackknife method is better than simple split  for estimating the score function.  However, in later experiments,  it seems that SRS has lower FDP than AutoMS.
Based on this, could you please adding another version AutoMS-based method in section 5.3 and 5.4, which is AutoMS-simple,  which just replace the Jackknife method with simple split?


Other minor comments:
page5 line 189:  ADNER --> AutoMS??

###########

update: Thanks for the author's response. I will keep my score as 6.


**Limitations:**

yes

**Strengths And Weaknesses:**

This paper is very clearly written and easy to understand. I really enjoy reading this paper and it makes interesting contribution.  The key idea is  estimating more "stable" p-value for better FDR and then adding extra step (i.e., model selection) for additional TDR control.  It's not surprising to see in experiments this method have better TDR than those methods without controlling TDR.

This paper can be seen a good extension work of Bates et al. [17]. The authors replace the simple split conformal prediction with Jackknife technique for more "accurate" estimated p-value,  by fully exploring the clean data and avoid the randomness caused by data-splitting. This idea is very straightforward.  Another contribution is selecting the best model from a pool of detectors. “Best" here means that the model detected the most outliers in the new dataset, which is not novel technique as well. Overall,  the novelty of this work is limited.  The manuscript will benefit from adding explanation about the novelty of such combination of two existing techniques, theoretically or practically.

---

> ### Author Response · Authors · 2022-08-02
> **Response to Reviewer 47FE**
>
> Q1: In page5 rationale, you explain why we can select the model which detects the most outliers as the "best" one. ... The issue may be that you assume that all the models will keep the FDR level constant.
>
> A1:
> - 1) It is indeed a conservative model A is better in your special case which has a strong signal where outliers are easy to be detected. But in more general cases where the signal is not strong enough, too aggressive models may detect more false discoveries, while too conservative models may miss weak signals/outliers. But actually, a smaller threshold could detect a larger number of discoveries including more false discoveries, which means the FDP and TDP change in the same direction. Our goal is to have a higher TDR while controlling the FDR under a target level. To achieve the goal, the FDP will tend to be closer to the preset level when obtaining a larger TDP. Therefore, the FDP of a competitive good detector will be roughly around our target level.
>
> - 2) The preset FDR level should be an acceptable level, which means all detectors that control the FDR below this level should be accepted. As long as the FDR of the selected detector is below a preset level, we have achieved our goal on FDR and focused on improving TDR.
>
> Q2: The main theoretical result is based on assumption 4.1-4.3. It is very common to require the assumption of learning stability and density. However, the assumption of rates seems a little complicated here. Could you give more explanation about this assumption?
>
> A2:
> - 1) In general, Assumption 4.3 is a technical one to ensure the uniform consistency of estimated p-values, which performs an important role of the convergence of the FDR control.
>
> - 2) Specifically, $A_n=o(n)$ is a pre-specified sequence and gives a non-trivial lower bound of true p-value function, $G_{\mathcal{M}}(t)\geq \frac{A_n}{n}=o(1)$. The $B_m$ is a quantity used to measure the difference magnitude of two p-values, e.g., the difference between the p-values based full sample score function and the Jackknife ones in Lemma C.3, or the difference between the joint p-values of pair $(j,k)$ based on the Jackknife score estimators and the leaving-$(j,k)$-out estimators in Lemma C.2. That's why we need $B_m=o(1)$, i.e two p-values should be close enough. This is a very mild condition and we can expect that the p-values will be close if the training sample sizes of inliers $m$ and test sample sizes of inliers $n_0$ are large enough in practice.
>
> - 3) Note that the number of detectors $\varpi_m$ is allowed to go to infinity with some rate for the proposed AutoMS. The assumptions $n\varpi_m B_m=o(1)$ and $m\varpi_mA_n^{-2/3}=o(1)$ indicates the required rate of $\varpi_m$ to ensure the uniform consistency among $\varpi_m$ detectors.
>
> Q3: In section 5.2, the results shows that the Jackknife method is better than simple split for estimating the score function. However, in later experiments, it seems that SRS has lower FDP than AutoMS. Based on this, could you please adding another version AutoMS-based method in section 5.3 and 5.4, which is AutoMS-simple, which just replace the Jackknife method with simple split?
>
> A3:
> - 1) SRS can be combined with our model selection procedure, hereafter called AutoMS-SRS. Experiment results on AutoMS-SRS and AutoMS-JK is showed in **General Response**. AutoMS-SRS can be regarded as a special case of AutoMS and also has the theoretical guarantees that the selected model yields **asymptotically** valid FDR control. But SRS does not fully explore the clean data and can cause randomness by data-splitting. So we use the Jackknife method instead of SRS to improve the accuracy and stability of the estimated p-values and enhance detection power.
>
> - 2) The reason why the FDR control of AutoMS is more difficult than SRS is that the FDP distribution of our selected detector is different from a given detector. Take a simple example: there are 10 standard Gaussian variables, each with a mean 0. The largest variable among those 10 does not have a mean 0. Therefore, the selected detector is more difficult to control FDR.
>
> - 3) As we explained in A1 (1), a smaller threshold could detect a larger number of discoveries including more false discoveries, which means the FDP and TDP change in the same direction. If we want a higher TDP, the FDP will tend to increase as well. From the Experiment results in **General Response**, both AutoMS-SRS and AutoMS-JK have this problem caused by the selection step. But we can see from Theorem 4.1 and the experimental results that the deviation of FDP is very small. Our procedure can select the detector that discovers more outliers while the FDP is still controlled **asymptotically around the FDR level**. Therefore, our AutoMS strategy can be more practical in real applications, which takes into account FDR and TDR and selects the best detector.

---

### Author Response · Authors · 2022-08-02
**General Response to All Reviewers**

Dear reviewers, we thank for your great efforts and valuable comments on our paper. Below, we would like to response your questions generally.

First, we would like to emphasize our contributions again.

- **1) We propose a criteria for model selection from the perspective of FDR control, which ideally finds a "best" detector with the largest TDR while controlling FDR.** Our AutoMS method can give theoretical guarantees for the TDP and FDR, while most existing model selection work lacks statistical theoretical guarantees.

- **2) With FDR control, the selected detector with the largest number of discoveries is roughly the one with the largest TDR.** Otherwise, without FDR control, just selecting the largest number of discoveries is not even a correct criteria, in this case, most detected discoveries are inliers and true outliers are not be detected. The rationale in lines 157-164 explain the rationality of our selection strategy.

- **3) We give a new theoretical guarantees that the FDR of the selected best detector can be controlled asymptotically below the target level.**
That is different from the one in Bates et al.(2021), which only focuses on the FDR control with one fixed detector under one time data-splitting.

- **4) We use the Jackknife method instead of the split-conformal approach in Bates et al (2021) to improve the accuracy and stability of the estimated p-value**, by fully exploring the clean data and avoiding the randomness caused by data-splitting.

**Comment: Experiment results on AutoMS-SRS and AutoMS-JK**

**Response:**
We appreciate the suggestion to add another experiment that combines our model selection procedure with SRS, hereafter called AutoMS-SRS. AutoMS with Jackknife is called AutoMS-JK. The table below shows results of AutoMS-JK and AutoMS-SRS when the target FDR level is 0.10. And we will add experiments with AutoMS-SRS in the final version.

Data set |  AutoMS-JK |  | AutoMS-SRS | |
------ | ------ |------ | ------ | ------
| |  FDP | TDP  |  FDP  | TDP
Covertype   | 0.106 | 0.933 | 0.105| 0.905
Credit Card | 0.099 | 0.758 | 0.104 | 0.757
Satellite      | 0.114 | 0.968 | 0.134 | 0.963
Shuttle        | 0.100 | 0.942 | 0.102 | 0.926

We see that the FDP of AutoMS-SRS is as large as that of AutoMS-JK, or even worse (the larger deviation than 0.1, the worse). Actually, the larger FDP is caused by the selection step. So AutoMS-SRS also has a larger FDP than SRS. But AutoMS-JK has a higher TDP than AutoMS-SRS.

---

### Meta-Review · Area_Chair_fFZ4 · 2022-08-29

**Recommendation:** Accept
**Confidence:** Less certain

**Metareview:**

The paper proposes a method for finding the best anomaly detector among a set of candidate methods that are all based on constructing a score function. The selection method is based on a leave-one-out estimate. Some theoretical results are presented and proven in the appendix, and in addition, some experiments are reported.

Overall, this paper presents a novel and interesting method for an important problem, and the theoretical considerations are certainly a plus. The only major issue of the paper is that only 4 real world data sets were considered, and despite the fact that this problem was raised by the reviewers, the authors did not include more during the rebuttal phase.
From my perspective, a strongly theoretical paper does not require extensive experiments, but the paper under review does not fall into this category. And for this reason, more experiments, on say another 15 data sets would have been really helpful.

In summary, this is an interesting paper with a sufficiently good theoretical part and some promising experiments. The latter could have been more, but overall this paper should be accepted.

**Award:**

No

---

### Decision · Program_Chairs · 2022-09-14

Accept